# Rethinking the Truly Unsupervised Image-to-Image Translation

## Abstract

Every recent image-to-image translation model uses either image-level (*i.e.* input-output pairs) or set-level (*i.e.* domain labels) supervision at a minimum. However, even the set-level supervision can be a serious bottleneck for data collection in practice. In this paper, we tackle image-to-image translation in a fully unsupervised setting, *i.e.*, neither paired images nor domain labels. To this end, we propose a truly unsupervised image-to-image translation model (TUNIT) that simultaneously learns to separate image domains and translate input images into the estimated domains. Experimental results show that our model achieves comparable or even better performance than the set-level supervised model trained with full labels, generalizes well on various datasets, and is robust against the choice of hyperparameters (*e.g.* the preset number of pseudo domains). In addition, TUNIT extends well to the semi-supervised scenario with various amount of labels provided.

## 1 Introduction

Given an image of one domain, image-to-image translation is a task to generate the plausible images of the other domains. Based on the success of conditional generative models (Mirza & Osindero, 2014; Sohn et al., 2015), many image translation methods have been proposed either using *image-level* supervision (*e.g.* paired data) (Isola et al., 2017; Hoffman et al., 2018; Zhu et al., 2017b; Wang et al., 2018; Park et al., 2019) or using *set-level* supervision (*e.g.* domain labels) (Zhu et al., 2017a; Kim et al., 2017; Liu et al., 2017; Huang et al., 2018; Liu et al., 2019; Lee et al., 2020). Though the latter approach is generally called *unsupervised* as a counterpart of the former, it actually assumes that the domain labels are given *a priori*. This assumption can be a serious bottleneck in practice as the number of domains and samples increases. For example, labeling individual samples of a large dataset, such as FFHQ, is expensive, and the distinction across domains can be ambiguous.

Here, we first clarify that unsupervised image-to-image translation should strictly denote the task *without any supervision* neither paired images nor domain labels. Under this definition, our goal is to develop an unsupervised translation model given a mixed set of images of many domains (Figure 1). We tackle this problem by formulating three sub-problems: 1) clustering the images by approximating the set-level characteristics (*i.e.* domains), 2) encoding the individual content and style of an input image, and 3) learning a mapping function among the estimated domains.

To this end, we introduce a `guiding network` that simultaneously solves 1) unsupervised domain classification and 2) style encoding. It has two branches of providing pseudo domain labels and encoding style features, which are later used in the discriminator and the generator training, respectively. We employ a differentiable clustering method based on mutual information maximization for estimating domain labels. This helps the guiding network group similar images together while evenly separate their categories. For embedding style codes, we adopt a contrastive loss (Hadsell et al., 2006; He et al., 2020; Chen et al., 2020a), which leads the model to further understand the dissimilarity between images, resulting in better representation learning. Finally, conditioned on the style features and domain labels from the guiding network, we use generative adversarial networks (GAN) to learn the image translation functions across various domains.

Although GAN and the guiding network play different roles, we do not separate their training process– our guiding network participates in the translation process. By doing so, the guiding network can exploit gradients from GAN training. The guiding network now understands the recipes of

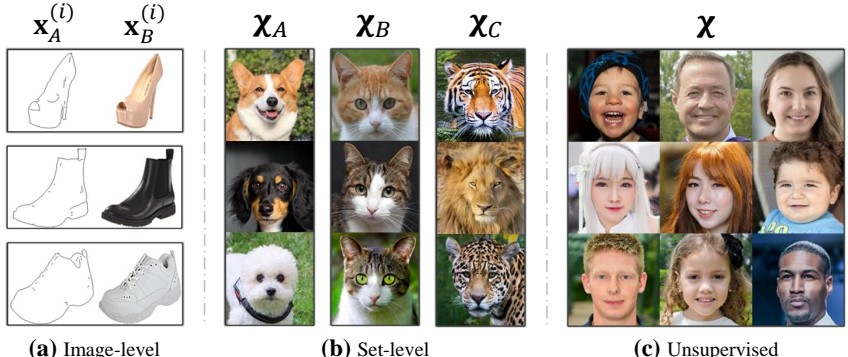

$$\mathbf{x}_A^{(i)} \quad \mathbf{x}_B^{(i)} \qquad \mathcal{X}_A \qquad \mathcal{X}_B \qquad \mathcal{X}_C \qquad\qquad \mathcal{X}$$

**(a)** Image-level      **(b)** Set-level      **(c)** Unsupervised

Figure 1: **Levels of supervision.** To perform image-to-image translation, existing methods need either **(a)** a dataset with input-output pairs or, **(b)** a dataset with domain information. Our method is capable of learning mappings among multiple domains using **(c)** a dataset without any supervision.

domain-separating attributes because the generator wants the style code to contain sufficient information to fool the domain-specific discriminator, and vice versa. Thanks to this interaction between the guiding network and GAN, our model successfully separates domains and translates images.

We quantitatively and qualitatively compare our model with the existing set-level supervised method under unsupervised and semi-supervised setting. The experiments on various datasets show that the proposed model outperforms the previous method over all different levels of supervision. Our experimental results show that, by exploiting the synergy between two tasks, the guiding network helps the image translation model to largely improve the generation performance.

Our contributions are summarized as follows:

- We clarify the definition of unsupervised image-to-image translation and to the best of our knowledge, our model is the first to succeed in this task in an end-to-end manner.
- We propose the guiding network to handle the unsupervised translation task and show that the interaction between translation and clustering is helpful for the task.
- We show the effectiveness of our model through the extensive experiments on various datasets.
- We confirm that our model is applicable to various numbers of clusters and the practical case, where ground truth labels of several samples are available.

## 2   TRULY UNSUPERVISED IMAGE-TO-IMAGE TRANSLATION (TUNIT)

We consider the unsupervised image-to-image translation problem, where we have images $\mathcal{X}$ from $K$ domains ($K \geq 2$) without domain labels $y$. Here, $K$ is an unknown property of the dataset. Throughout the paper, we denote $K$ as the actual number of domains in a dataset and $\hat{K}$ as the arbitrarily chosen number of domains to train models. We design a module that integrates both a domain classifier and a style encoder, which we call guiding network. It guides the translation by feeding reference images as the style code to the generator and as the pseudo domain labels to the discriminator. Using the feedback from the discriminator regarding the pseudo labels, the generator synthesizes images of the target domains (*e.g.* breeds) while respecting styles (*e.g.* fur patterns) of the reference images and maintaining the content (*e.g.* pose) of source images (Figure 2).

### 2.1   LEARNING TO PRODUCE DOMAIN LABELS AND ENCODE STYLE FEATURES

In our framework, the guiding network $E$ plays a central role as an unsupervised domain classifier as well as a style encoder. Our guiding network $E$ consists of two branches, $E_C$ and $E_S$, each of which learns to provide domain labels and style codes, respectively. In experiments, we compare our guiding network against straightforward approaches, *i.e..*, K-means on image or feature space.

**Unsupervised domain classification.** The discriminator requires target domain labels to provide useful gradients for image translation into the target domain. $E_C$ adopts a differentiable clustering

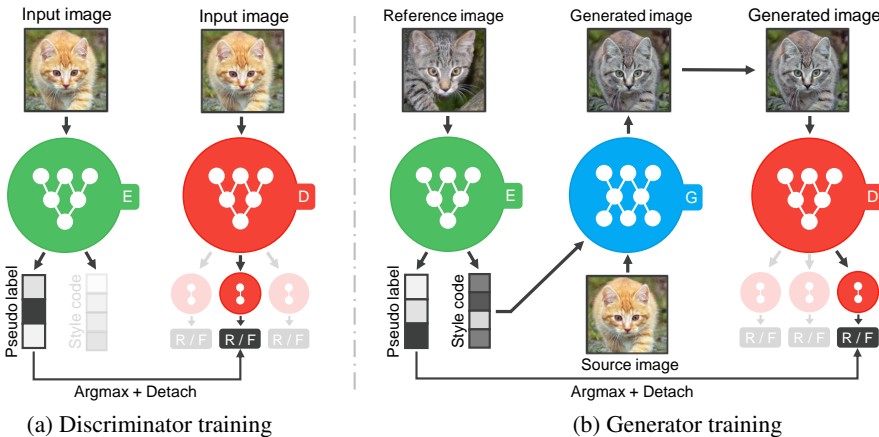

| (a) Discriminator training | (b) Generator training |

Figure 2: **Overview of our proposed method.** The figure illustrates how our model changes the breed of the cat. **(a)** An estimated domain from our guiding network $E$ is used to train the multi-task discriminator $D$. **(b)** $E$ provides the generator $G$ with the style code of a reference image and the estimated domain is again used for GAN training.

technique to provide pseudo domain labels of reference images, maximizing the mutual information (MI) between an image $\mathbf{x}$ and its randomly augmented version $\mathbf{x}^+$ (Ji et al., 2019). The optimum of the mutual information $I(\mathbf{p}, \mathbf{p}^+)$ is reached as the entropy $H(\mathbf{p})$ is maximum and the conditional entropy $H(\mathbf{p}|\mathbf{p}^+)$ is minimum, where $\mathbf{p} = E_C(\mathbf{x})$ represents the softmax output from $E_C$, indicating a probability vector of $\mathbf{x}$ over $\hat{K}$ domains. Please refer to Section 3.3 for more details about $\hat{K}$. Maximizing MI encourages $E_C$ to assign the same domain label to the pair ($\mathbf{x}$ and $\mathbf{x}^+$) while evenly distributing entire samples to all domains.

Formally, $E_C$ maximizes the mutual information:

$$\mathcal{L}_{MI} = I(\mathbf{p}, \mathbf{p}^+) = I(\mathbf{P}) = \sum_{i=1}^{\hat{K}} \sum_{j=1}^{\hat{K}} \mathbf{P}_{ij} \ln \frac{\mathbf{P}_{ij}}{\mathbf{P}_i \mathbf{P}_j}, \tag{1}$$

$$\text{where } \mathbf{P} = \mathbb{E}_{\mathbf{x}^+ \sim f(\mathbf{x})|\mathbf{x} \sim p_{data}(\mathbf{x})}[E_C(\mathbf{x}) \cdot E_C(\mathbf{x}^+)^T],$$

where $f$ is a composition of random augmentations such as random cropping and affine transformation. $\mathbf{P}_i = \mathbf{P}(\mathbf{p} = i)$ denotes the $\hat{K}$-dimensional marginal probability vector, and $\mathbf{P}_{ij} = \mathbf{P}(\mathbf{p} = i, \mathbf{p}^+ = j)$ denotes the joint probability. To provide a deterministic one-hot label to the discriminator, we use the $\texttt{argmax}$ operation (i.e. $y = \texttt{argmax}(E_C(\mathbf{x}))$). We note that the mutual information is one way to implement TUNIT, therefore, any differentiable clustering methods can be adopted such as SCAN (Van Gansbeke et al., 2020).

**Style encoding and improving domain classification.** $E_S$ encodes an image into a style code $\mathbf{s}$ which provides translation guide for the generator. In addition to the style guide to the generator, $E_S$ is beneficial in improving unsupervised domain classification, where pseudo labels from $E_C$ fail to scale up when samples are complex and diverse (*e.g.*, AnimalFaces (Liu et al., 2019)). Since $E_S$ is an another branch of the guiding network, imposing the contrastive loss (He et al., 2020) on the style codes improves representation of the shared embeddings:

$$\mathcal{L}_{style}^E = -\log \frac{\exp(\mathbf{s} \cdot \mathbf{s}^+/\tau)}{\sum_{i=0}^N \exp(\mathbf{s} \cdot \mathbf{s}_i^-/\tau)}, \tag{2}$$

where $\mathbf{s} = E_S(\mathbf{x})$. $\mathbf{x}$ and $\mathbf{x}^+$ denote an image and randomly augmented version of $\mathbf{x}$, respectively. This $(N + 1)$-way classification enables $E$ to utilize not only the similarity of the positive pair ($\mathbf{s}$, $\mathbf{s}^+$) but also the dissimilarity of the negative pairs ($\mathbf{s}$, $\mathbf{s}_i^-$). We adopt a queue to store the negative codes $\mathbf{s}_i^-$ of the previously sampled images as MoCo (He et al., 2020). By doing so, we can conduct the contrastive learning efficiently without large batch sizes (Saunshi et al., 2019). We observe that adding this objective significantly improves unsupervised classification accuracy on AnimalFaces from 68.0% to 84.1% compared to the previous approach (Ji et al., 2019).

## 2.2 LEARNING TO TRANSLATE IMAGES WITH THE DOMAIN GUIDANCE

In this subsection, we describe how to perform the unsupervised image-to-image translation under the guidance of our guiding network. For successful translation, the model should provide the realistic images containing the visual feature of the target domain. To this end, we adopt three losses: 1) adversarial loss to produce realistic images, 2) style contrastive loss that encourages the model not to ignore the style codes, 3) image reconstruction loss for preserving the domain-invariant features. We explain each loss and the overall objective for each network.

**Adversarial loss.** For adversarial training, we adopt a variant of conditional discriminator, the multi-task discriminator (Mescheder et al., 2018). It is designed to conduct discrimination for each domain simultaneously. However, its gradient is calculated only with the loss for estimating the domain of the input image. For the domain label of the input image, we utilize the pseudo label from the guiding network. Formally, given the pseudo label $\tilde{y}$ for a reference image $\tilde{\mathbf{x}}$, we train our generator $G$ and multi-task discriminator $D$ via the adversarial loss:

$$\mathcal{L}_{adv} = \mathbb{E}_{\tilde{\mathbf{x}} \sim p_{data}(\mathbf{x})}[\log D_{\tilde{y}}(\tilde{\mathbf{x}})] + \mathbb{E}_{\mathbf{x}, \tilde{\mathbf{x}} \sim p_{data}(\mathbf{x})}[\log(1 - D_{\tilde{y}}(G(\mathbf{x}, \tilde{\mathbf{s}})))], \qquad (3)$$

where $D_{\tilde{y}}(\cdot)$ denotes the logit from the domain-specific ($\tilde{y}$) discriminator, and $\tilde{\mathbf{s}} = E_S(\tilde{\mathbf{x}})$ denotes a target style code of the reference image $\tilde{\mathbf{x}}$. The generator $G$ learns to translate $\mathbf{x}$ to the target domain $\tilde{y}$ while reflecting the style code $\tilde{\mathbf{s}}$.

**Style constrastive loss.** In order to prevent a degenerate case where the generator ignores the given style code $\tilde{\mathbf{s}}$ and synthesize a random image of the domain $\tilde{y}$, we impose a style contrastive loss:

$$\mathcal{L}_{style}^G = \mathbb{E}_{\mathbf{x}, \tilde{\mathbf{x}} \sim p_{data}(\mathbf{x})} \left[ -\log \frac{\exp(\mathbf{s}' \cdot \tilde{\mathbf{s}})}{\sum_{i=0}^{N} \exp(\mathbf{s}' \cdot \mathbf{s}_i^- / \tau)} \right]. \qquad (4)$$

Here, $\mathbf{s}' = E_S(G(\mathbf{x}, \tilde{\mathbf{s}}))$ denotes the style code of the translated image $G(\mathbf{x}, \tilde{\mathbf{s}})$ and $\mathbf{s}_i^-$ denotes the negative style codes, which are from the same queue used in equation (2). And we follow the training scheme of MoCo (He et al., 2020) as equation (2). The above loss guides the generated image $G(\mathbf{x}, \tilde{\mathbf{s}})$ to have a style similar to the reference image $\tilde{\mathbf{x}}$ and dissimilar to negative (other) samples. By doing so, we also avoid the degenerated solution where the encoder maps all the images to the same style code of the reconstruction loss (Choi et al., 2020) based on L1 or L2 norm. Equation (2) and (4) are based on contrastive loss, but they are used for different purposes. Please refer to Appendix H for more discussion.

**Image reconstruction loss.** To ensure that the generator $G$ can reconstruct the source image $\mathbf{x}$ when given with its original style $\mathbf{s} = E_S(\mathbf{x})$, we impose an image reconstruction loss:

$$\mathcal{L}_{rec} = \mathbb{E}_{\mathbf{x} \sim p_{data}(\mathbf{x})}[\|\mathbf{x} - G(\mathbf{x}, \mathbf{s})\|_1]. \qquad (5)$$

This objective not only ensures the generator $G$ to preserve domain-invariant characteristics (e.g., pose) of its input image $\mathbf{x}$, but also helps to learn the style representation of the guiding network $E$ by extracting the original style $\mathbf{s}$ of the source image $\mathbf{x}$.

**Overall objective.** Finally, we train the three networks jointly as follows:

$$\begin{aligned} \mathcal{L}_D &= -\mathcal{L}_{adv}, \\ \mathcal{L}_G &= \mathcal{L}_{adv} + \lambda_{style}^G \mathcal{L}_{style}^G + \lambda_{rec} \mathcal{L}_{rec}, \\ \mathcal{L}_E &= \mathcal{L}_G - \lambda_{MI} \mathcal{L}_{MI} + \lambda_{style}^E \mathcal{L}_{style}^E \end{aligned} \qquad (6)$$

where $\lambda$'s are hyperparameters. Note that our guiding network $E$ receives feedback from $L_G$, which is essential for our method. We discuss the effect of feedback to $E$ on performance in Section 3.1.

## 3 EXPERIMENTS

We first evaluate TUNIT on labeled datasets by treating them as unlabeled because the desired behaviours of the translation models in labeled datasets are well defined (Section 3.1). Here, we provide an ablation study to analyze the effect of each component and compare the models both quantitatively and qualitatively. We then move on to unlabeled datasets to validate our model in the unsupervised scenario in the wild (Section 3.2). Lastly, we show that TUNIT is robust against the choice

| | Configuration | AnimalFaces-10 | | | Food-10 | | |
|---|---|---|---|---|---|---|---|
| | | mFID | D & C | Acc. | mFID | D & C | Acc. |
| A | Baseline FUNIT (supervised) | 74.0 | 0.749 / 0.671 | **1.000** | 68.4 | 0.989 / 0.782 | **1.000** |
| B | (A) + Improved G & D (supervised) | **46.2** | **0.896 / 0.732** | **1.000** | **57.6** | **1.284 / 0.857** | **1.000** |
| C | (B) + K-means on image space | 110.7 | 0.822 / 0.615 | 0.215 | 90.7 | 0.849 / 0.648 | 0.201 |
| D | (B) + K-means on feature space | 76.2 | 0.770 / 0.597 | 0.428 | 64.6 | 0.968 / 0.808 | 0.331 |
| E | (B) + Differentiable clustering | 73.5 | 0.940 / 0.588 | 0.680 | 64.2 | 1.038 / 0.819 | 0.542 |
| F | TUNIT w/ sequential training | **46.0** | 1.060 / 0.789 | **0.850** | 61.1 | 0.908 / 0.777 | **0.860** |
| G | TUNIT w/ joint training | 47.7 | **1.039 / 0.805** | 0.841 | **52.2** | **1.079 / 0.875** | 0.848 |

Table 1: Main results. mFID, Density / Coverage (D & C), and classification accuracy (Acc) of each training configuration. Note that the configurations (A) - (B) use ground-truth class labels, while (C) - (G) use pseudo-labels. We **bold** the best results separately for supervised and unsupervised settings.

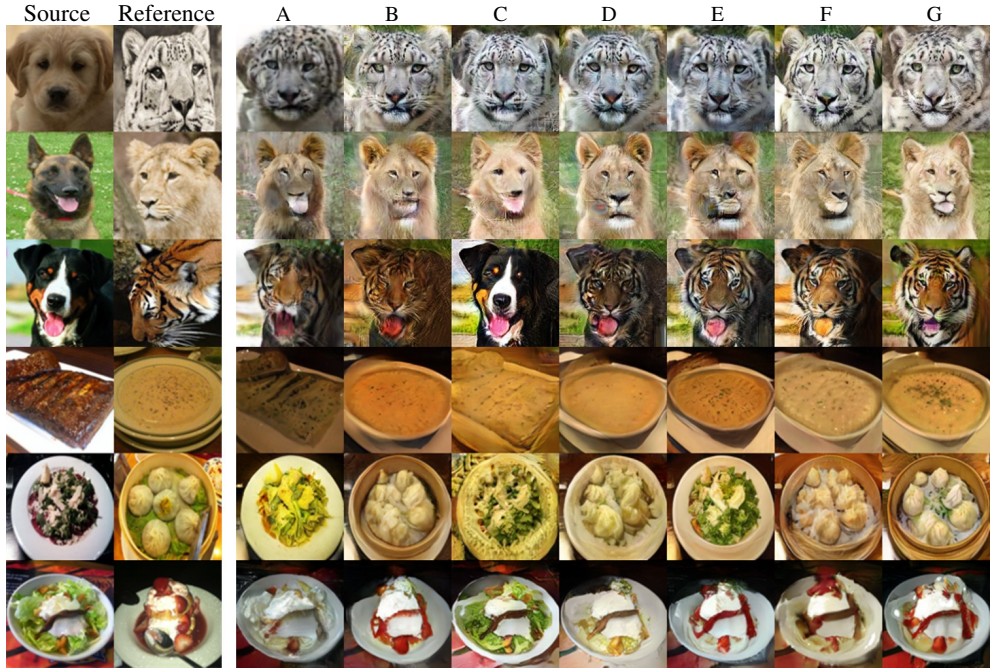

Figure 3: Qualitative comparison of translation results using each configuration in Table 1. Here, B reflects the style feature (*e.g.* species or type of food) of the reference images while A does not. The model C performs much worse than A and B in that it overly adopts the source image, not adequately merging styles and contents from both sides. The model D generates more plausible images than C but fails to reflect the characteristics of the reference images. For example, D on fifth row does not look like *several pieces of dumpling* due to its shape and dish color, meaning that the reference styles are not properly reflected. Similarly, E also fails to generate the dumpling in the fifth row. TUNIT with sequential training F reflects the visual features of each reference on both datasets. However, in terms of visual fidelity, we observe that G consistently outperforms F. Akin to the quantitative results, TUNIT achieves equivalent or even better visual quality than the set-level supervised model A and B.

of hyperparameters (e.g. the preset number of clusters, $K$) and extends well to the semi-supervised scenario (Section 3.3). In all experiments, we use FUNIT (Liu et al., 2019) as our baseline.

**Datasets.** For the labeled datasets, we randomly select ten classes among 149 classes of Animal-Faces and 101 classes of Food-101, which we call AnimalFaces-10 and Food-10, respectively. Here, the labels are used only for the evaluation purpose. For the unlabeled datasets, we use AFHQ, FFHQ, and LSUN Car (Choi et al., 2020; Karras et al., 2019; Yu et al., 2015), which do not have any or are missing with fine-grained labels. Specifically, AFHQ roughly has three groups (i.e., dog, cat and wild), but each group contains diverse species and these species labels are not provided. FFHQ and LSUN Car contain various human faces and cars without any labels, respectively.

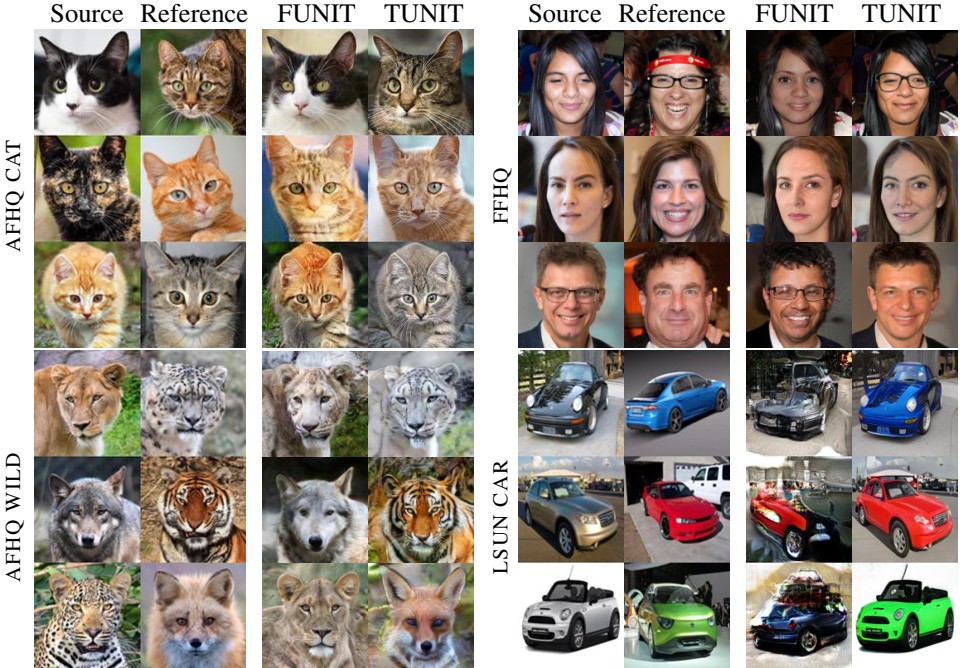

Figure 4: Reference-guided image translation results on unlabeled datasets.

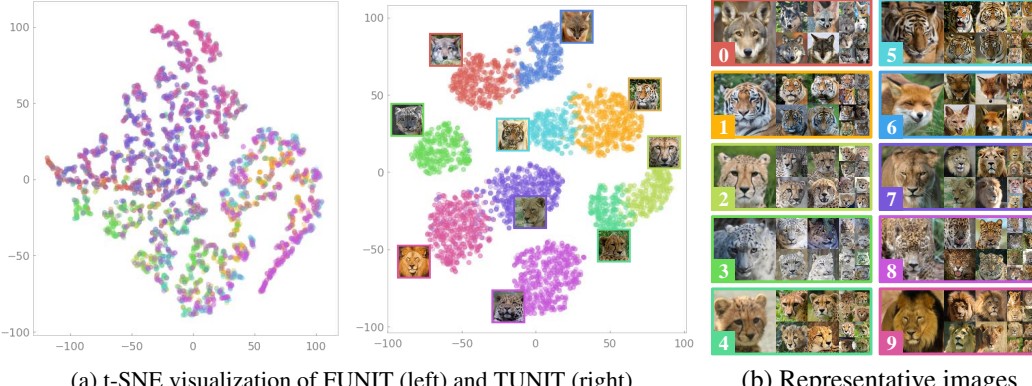

(a) t-SNE visualization of FUNIT (left) and TUNIT (right)  (b) Representative images

Figure 5: t-SNE visualization of the style space of our guiding network trained on AFHQ Wild. Since AFHQ Wild does not have ground-truth labels, each point is colored with the guiding network's prediction. Although we set the number of domains to be quite large ($\hat{K} = 10$), the network separates one species into two domains, which are so closely located that the model creates six clusters.

**Evaluation Metrics.** We report two scores to assess the generated images. First, to provide a general sense of image quality, we use the mean of class-wise Fréchet Inception Distance (mFID) (Heusel et al., 2017). It can avoid the degenerate case of the original FID, which assigns a good score when the model conveying the source image as is. Additionally, to provide a finer assessment of the generated images, we report Density and Coverage (D&C) (Naeem et al., 2020). D&C separately evaluates the fidelity and the diversity of the model outputs, which is also known to be robust against outliers and model hyperparameters (e.g. the number of samples used for evaluation). Denote that a lower mFID score means better image quality, and D&C scores that are bigger or closer to 1.0 indicate the better fidelity and diversity, respectively. Please refer to Appendix C for the detailed information.

### 3.1 COMPARATIVE EVALUATION ON LABELED DATASETS

Table 1 summarizes the effect of each component of TUNIT and rigorous comparisons with the state-of-the-art supervised method, FUNIT. First, we report the set-level supervised performance

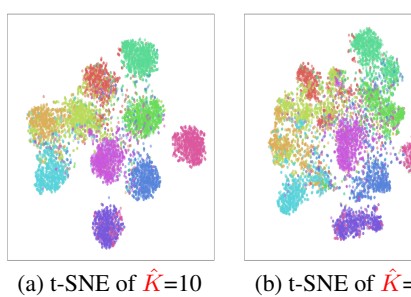

| $\hat{K}$ | AnimalFaces-10 | | Food-10 | |
|---|---|---|---|---|
| | mFID | D & C | mFID | D & C |
| 1 | 129.6 | 0.561 / 0.512 | 95.1 | **1.113** / 0.771 |
| 4 | 77.7 | 0.879 / 0.738 | 67.4 | 0.851 / 0.785 |
| 7 | 62.7 | 1.016 / 0.729 | 52.7 | 1.079 / **0.875** |
| 10 | **47.7** | 1.039 / **0.805** | **52.2** | 1.079 / **0.875** |
| 13 | 56.8 | 0.993 / **0.805** | 54.8 | 0.970 / 0.845 |
| 16 | 54.1 | **1.093** / 0.782 | 54.8 | 1.029 / 0.857 |
| 20 | 55.4 | 1.019 / 0.778 | 57.7 | 0.937 / 0.846 |
| 50 | 63.8 | 0.858 / 0.701 | 60.8 | 1.067 / 0.837 |
| 500 | 67.2 | 0.921 / 0.694 | 63.2 | 0.986 / 0.826 |
| 1000 | 66.9 | 0.908 / 0.707 | 60.7 | 0.945 / 0.845 |

(a) t-SNE of $\hat{K}$=10     (b) t-SNE of $\hat{K}$=20

Table 2: t-SNE visualization of the model with (a) $\hat{K}$=10 and (b) $\hat{K}$=20 trained on AnimalFaces-10 and quantitative evaluation of our method by varying the number of pseudo domains $\hat{K}$. Each point is colored with the ground-truth labels. As shown in t-SNE visualizations, even if $\hat{K}$ is set to overly larger than the actual number of domains, the guiding network clusters the domains reasonably well.

Source   Reference   $\hat{K}$=1   $\hat{K}$=4   $\hat{K}$=7   $\hat{K}$=10   $\hat{K}$=13   $\hat{K}$=16   $\hat{K}$=20

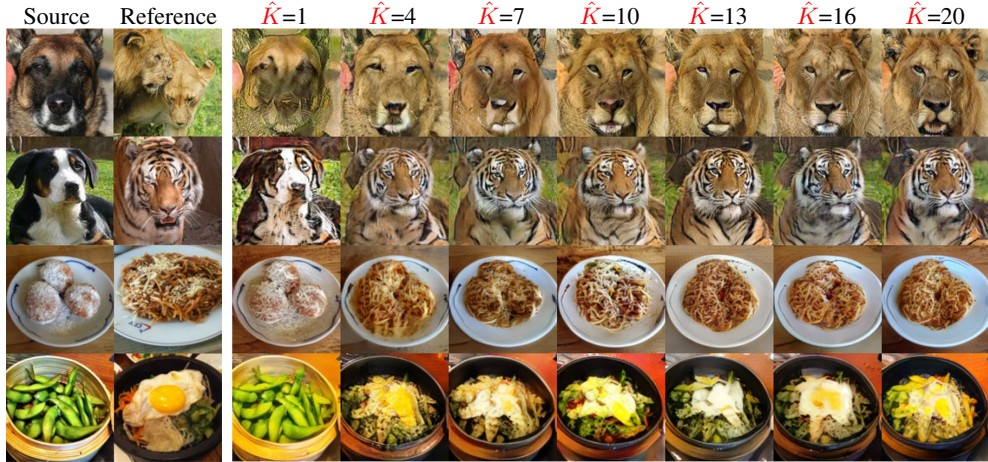

Figure 6: Qualitative comparison on the number of pseudo domains $\hat{K}$. The performance varies along with $\hat{K}$. When we set $\hat{K}$ large enough, the results are reasonable.

of FUNIT and its variant (Table 1). Here, A is the original FUNIT and B denotes the modified FUNIT using our architecture (*e.g.* We do not use PatchGAN discriminator), which brings a large improvement over every score on both datasets. One simple way to extend B to the unsupervised scenario is to add an off-the-shelf clustering method and use its estimated labels instead of the ground truth. We employ K-means clustering on the image space for C, and the pretrained feature space for D. Here, we use ResNet-50 (He et al., 2016) features trained with MoCo v2 (Chen et al., 2020b) on ImageNet. Not surprisingly, because the estimated labels are inaccurate, the overall performance significantly drops. Although using the pretrained features helps a little, not only is it far from the set-level supervised performance but it requires three steps to train the entire model, which complicates the application. This can be partially addressed by employing the differentiable clustering method (Ji et al., 2019), which trains VGG-11BN (Simonyan & Zisserman, 2015) with mutual information maximization that makes E. This reduces the number of training steps from three to two and provides better label estimation, which enables the model to approach the performance of original FUNIT A. However, as seen in the coverage score, the sample diversity is unsatisfactory.

Finally, we build TUNIT by introducing the guiding network and the new objective functions described in Section 2. The changes significantly improve the accuracy on both datasets, particularly achieving similar mFID of the improved set-level supervised model B. Our final model, G matches or outperforms mFID and D&C of B. This is impressive because B utilizes oracles for training while G has no labels. Notably, TUNIT can cover 7%p wider support of the data on AnimalFaces-10 than B. We conjecture that TUNIT benefits from the style codes that represent meaningful domain features learned by clustering. By comparing F and G, we confirm that they are comparable in terms of clustering and G is more stable in terms of inter-dataset performance. Therefore, we adopt the joint

| Configuration | AnimalFaces-10 | | | | Food-10 | | | |
|---|---|---|---|---|---|---|---|---|
| | 20% | 40% | 60% | 80% | 20% | 40% | 60% | 80% |
| A  FUNIT | 124.4 | 106.4 | 96.0 | 79.6 | 111.4 | 85.8 | 74.8 | 70.3 |
| G  TUNIT (ours) | **42.0** | **42.6** | **43.9** | **46.2** | **53.6** | **56.2** | **52.8** | **53.4** |

Table 3: Quantitative evaluation (mFID) when few labels are available during training.

| Configuration | AnimalFaces-10 | | | | Food-10 | | | |
|---|---|---|---|---|---|---|---|---|
| | 1% | 2% | 4% | 8% | 1% | 2% | 4% | 8% |
| A  FUNIT | 107.8 | 104.7 | 90.3 | 93.9 | 71.9 | 71.5 | 71.6 | 69.0 |
| G  TUNIT (ours) | **47.9** | **44.8** | **42.7** | **42.4** | **54.5** | **56.1** | **55.9** | **55.8** |

Table 4: Quantitative evaluation (mFID) when few labels are available during training. Here, an auxliary classifier is adopted to improve the FUNIT baseline by giving pseudo-labels to $\mathcal{D}_{un}$.

training of style encoder and clustering as our final model (G). In addition, we remove the adversarial loss for training the guiding network. It directly degrades the performance; mFID changes from 47.7 to 63.0 on AnimalFaces-10. It indicates that our training scheme takes an important portion of performance gains. Qualitative results also show superiority of TUNIT over competitors (Figure 3).

## 3.2  VALIDATION ON UNLABELED DATASET

We evaluate TUNIT on the unlabeled datasets having no clear separations of the domains, which are AFHQ, FFHQ and LSUN Cars. For AFHQ, we train three individual models for *dog*, *cat* and *wild*. For all experiments, FUNIT is used as a baseline. We train it by presuming all labels to be the same as one. We set $\hat{K}$=10 for all the TUNIT models. More discussions on $\hat{K}$ will be in Section 3.3.

Figure 4 demonstrates the results. We observe that the results of TUNIT adequately reflect the style feature of the references such as the textures of cats or cars and the species of the wilds. Although FFHQ has no clear domain distinctions, TUNIT captures the existence of glasses or smile as domains, and then add or remove glasses or smile. However, FUNIT performs much worse than TUNIT in this truly unsupervised scenario. For example, FUNIT outputs the inputs as is (cats and wilds) or insufficiently reflects the species (third row of AFHQ Wild). For FFHQ, despite that FUNIT makes some changes, the changes are not interpreted as meaningful domain translations. For LSUN Car, FUNIT fails to keep the fidelity.

We also visualize the style space of both models to qualitatively assess the quality of the representation. Figure 5 shows the t-SNE maps trained on AFHQ Wild and the examples of each cluster. Surprisingly, TUNIT organizes the samples according to the species where it roughly separates the images into six species. Although we set $\hat{K}$ to be overly large, the model represents one species into two domains where those two domains position much closely. The highly disentangled, meaningful style features can be an important factor in the success of our model. On the other hand, the style features of FUNIT hardly learn meaningful domains so that the model cannot conduct the translation properly as shown in Figure 4. Because of the page limit, we include more results in Appendix E, G.

## 3.3  ANALYSIS ON GENERALIZABILITY

**Robustness to various $\hat{K}$'s.** When TUNIT conducts clustering for estimating domain labels, the number of clusters $\hat{K}$ can affect the overall performances. Here, we study the effects on different $\hat{K}$ on the labeled datasets and report them in Figure 6 and Table 2. As expected, the model performs best in terms of mFID when $\hat{K}$ equals to the ground truth $K$ (*i.e.* $\hat{K}$=10). Additionally, TUNIT performs reasonably well for sufficiently large $\hat{K}$ ($\geq$ 7), and even with 100 times larger $\hat{K}$ than the actual number of the domains, TUNIT still works well on both datasets. From this study, we conclude that TUNIT is relatively robust against $\hat{K}$ as long as it is sufficiently large. We suggest using a sufficiently large number of $\hat{K}$ or studying different $\hat{K}$'s in log scale to find the optimal model.

**With Few labels.** We investigate whether or not TUNIT is effective for the partially labeled dataset that corresponds to a more practical scenario. To this end, we use AnimalFaces-10 and Food-10. We partition the dataset $\mathcal{D}$ into the labeled set $\mathcal{D}_{sup}$ and the unlabeled set $\mathcal{D}_{un}$ with varying ratio $\gamma = |\mathcal{D}_{sup}|/|\mathcal{D}|$. Like before, we choose FUNIT as a competitor. We first train the networks while

changing $\gamma$ from 0.2 to 1.0 and report the result Table 3. As $\gamma$ decreases, the performance of FUNIT significantly degrades whereas our model maintains mFID around 45 and 55 on both datasets. We also train an auxiliary classifier (VGG-11BN) with $\mathcal{D}_{sup}$ then provide pseudo-labels for $\mathcal{D}_{un}$ to FUNIT. As shown in Table 4, the performance of FUNIT is no longer sensitive to the changes in $\gamma$ but still much worse than TUNIT. Although semi-supervised learning schemes can further improve FUNIT, TUNIT outperforms FUNIT using all labels as seen in Table 1. Under the empirical results in the semi-supervised scenario, we verify that TUNIT is also effective for semi-supervised setting.

## 4 CONCLUSION

We argue that the unsupervised image-to-image translation should not utilize any supervision, such as image-level (*i.e.* paired) or set-level (*i.e.* unpaired) supervision. Under this rigorous regime, many previous studies fall into the set-level supervised framework that uses the domain information at a minimum. In this paper, for the first time, we proposed an effective model to handle the truly unsupervised image-to-image translation. To this end, we suggested the guiding network that performs unsupervised representation learning for providing pseudo labels and the image translation tasks. The experimental results showed that guiding network indeed exploits the synergy between two tasks, and the proposed model successfully conducts the unsupervised-image-to-image translation. We also showed the generalizability on the value of $\hat{K}$ and the partially labeled scenario.

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

APPENDIX

## A    RELATED WORK

**Image-to-image translation.** Since the seminal work of Pix2Pix (Isola et al., 2017), image-to-image translation models have shown impressive results (Zhu et al., 2017a; Liu et al., 2017; Kim et al., 2017; Kupyn et al., 2018; Choi et al., 2018; Huang et al., 2018; Liu et al., 2019; Yang et al., 2019). Exploiting the cycle consistency constraint, these methods were able to train the model with a set-level supervision (domains) solely. However, acquiring domain information can be a huge burden in practical applications where a large amount of data are gathered from several mixed domains, *e.g.*, web images (Yalniz et al., 2019). Not only does this complicates the data collection, but it restricts the methods only applicable to the existing dataset and domains. Inspired from few shot learning, Liu et al. (2019) proposed FUNIT that works on previously unseen target classes. However, FU-NIT still requires the labels for training. Recently, Bahng et al. (Bahng et al., 2020) has partially addressed this by adopting the ImageNet pre-trained classifier for extracting domain information. Unlike the previous methods, we aim to design an image-to-image translation model that can be applied without any supervision such as a pre-trained network or supervision on both the train and the test datasets.

**Unsupervised representation learning and clustering.** Unsupervised representation learning aims to extract meaningful features for downstream tasks without any human supervision. To this end, many researchers have proposed to utilize the information that can be acquired from the data itself (Gidaris et al., 2018; Hjelm et al., 2019; Ji et al., 2019; He et al., 2020; Van Gansbeke et al., 2020). Recently, by incorporating the contrastive learning into a dictionary learning framework, MoCo (He et al., 2020) has achieved outstanding performance in various downstream tasks under reasonable mini-batch size. On the other hand, IIC (Ji et al., 2019) have utilized the mutual information maximization in a unsupervised way so that the network clusters images while assigning the images evenly. Though IIC provided a principled way to perform unsupervised clustering, it fails to scale up when combined with a difficult downstream task such as image-to-image translation. By taking the best of both worlds, we aim to solve unsupervised image-to-image translation.

## B    TRAINING DETAILS

We train the guiding network for the first 65K iterations while freezing the update from both the generator and the discriminator. Then, we train the whole framework 100K more iterations for training all the networks. The batch size is set to 32 and 16 for $128 \times 128$ and $256 \times 256$ images, respectively. Training takes about 36 hours on a single Tesla V100 GPU with our implementation using Py-Torch(Paszke et al., 2017). We use Adam (Kingma & Ba, 2014) optimizer with $\beta_1 = 0.9, \beta_2 = 0.99$ for the guiding network, and RMSprop (Hinton et al., 2012) optimizer with $\alpha = 0.99$ for the generator and the discriminator. All learning rates are set to 0.0001 with a weight decay 0.0001. We adopt hinge version adversarial loss (Lim & Ye, 2017; Tran et al., 2017) with $R_1$ regularization (Mescheder et al., 2018) using $\gamma = 10$ (Eq. 5). We set $\lambda_{\text{rec}} = 0.1, \lambda_{\text{style}}^G = 0.01, \lambda_{\text{style}}^E = 1$, and $\lambda_{\text{MI}} = 5$ in equation. 6 for all experiments. When the guiding network is simultaneously trained with the generator, we decrease $\lambda_{\text{style}}^E$ and $\lambda_{\text{MI}}$ to 0.1 and 0.5, respectively. For evaluation, we use the exponential moving average over the parameters (Karras et al., 2018) of the guiding network and the generator. We initialize the weights of convolution layers with He initialization (He et al., 2015), all biases to zero, and weights of linear layers from $N(0, 0.01)$ with zero biases. The source code will be available publicly.

## C    EVALUATION PROTOCOL

For evaluation, we use class-wise Fréchet Inception Distance (FID) (Heusel et al., 2017), which is often called mFID in literatures and D&C (Naeem et al., 2020). FID measures Fréchet distance between real and fake samples embedded by the last average pooling layer of Inception-V3 pre-trained on ImageNet. Class-wise FID is obtained by averaging the FIDs of individual classes. In the experiments with fewer labels, we report the mean value of best five mFID's over 100K iterations. For example, we use entire real images of each class and generate 810 fake images where $18 \times$

$(K-1)$ source images ($K = 10$ for AnimalFaces-10) and five reference images of AnimalFaces-10 are used to produce those fake images. We choose the source images from all classes except for the target class. For each source image, the five references are selected arbitrarily. For D&C, we generate fake images the similar number of training images with randomly selected source and reference images. Then, we use Inception-V3 pre-trained on ImageNet for extracting feature vectors and measure D&C by using the feature vectors.

## D ARCHITECTURE DETAILS

For the guiding network, we use `VGG11` before the linear layers followed by the average pooling operation as the shared part and append two branches $E_{\text{class}}$ and $E_{\text{style}}$. The branches are one linear layer with $\hat{K}$ and 128 dimensional outputs, respectively. The detailed information of the generator, the guiding network and the discriminator architectures are provided in Table 5, Table 6 and Table 7.

| LAYER | RESAMPLE | NORM | OUTPUT SHAPE |
|---|---|---|---|
| Image $\mathbf{x}$ | - | - | $128 \times 128 \times 3$ |
| Conv7×7 | - | IN | $128 \times 128 \times ch$ |
| Conv4×4 | Stride 2 | IN | $64 \times 64 \times 2ch$ |
| Conv4×4 | Stride 2 | IN | $32 \times 32 \times 4ch$ |
| Conv4×4 | Stride 2 | IN | $16 \times 16 \times 8ch$ |
| ResBlk | - | IN | $16 \times 16 \times 8ch$ |
| ResBlk | - | IN | $16 \times 16 \times 8ch$ |
| ResBlk | - | AdaIN | $16 \times 16 \times 8ch$ |
| ResBlk | - | AdaIN | $16 \times 16 \times 8ch$ |
| Conv5×5 | Upsample | AdaIN | $32 \times 32 \times 4ch$ |
| Conv5×5 | Upsample | AdaIN | $64 \times 64 \times 2ch$ |
| Conv5×5 | Upsample | AdaIN | $128 \times 128 \times ch$ |
| Conv7×7 | - | - | $128 \times 128 \times 3$ |

Table 5: Generator architecture. "ch" represents the channel multiplier that is set to 64. IN and AdaIN indicate instance normalization and adaptive instance normalization, respectively.

| LAYER | RESAMPLE | NORM | OUTPUT SHAPE |
|---|---|---|---|
| Image $\mathbf{x}$ | - | - | $128 \times 128 \times 3$ |
| Conv3×3 | MaxPool | BN | $64 \times 64 \times ch$ |
| Conv3×3 | MaxPool | BN | $32 \times 32 \times 2ch$ |
| Conv3×3 | - | BN | $32 \times 32 \times 4ch$ |
| Conv3×3 | MaxPool | BN | $16 \times 16 \times 4ch$ |
| Conv3×3 | - | BN | $16 \times 16 \times 8ch$ |
| Conv3×3 | MaxPool | BN | $8 \times 8 \times 8ch$ |
| Conv3×3 | - | BN | $8 \times 8 \times 8ch$ |
| Conv3×3 | MaxPool | BN | $4 \times 4 \times 8ch$ |
| GAP | - | - | $1 \times 1 \times 8ch$ |
| FC | - | - | 128 |
| FC | - | - | $\hat{K}$ |

Table 6: Guiding network architecture. "ch" represents the channel multiplier that is set to 64. The architecture is based on VGG11-BN. GAP and FC denote global average polling (Lin et al., 2013) and fully connected layer, respectively.

| LAYER | RESAMPLE | NORM | OUTPUT SHAPE |
|---|---|---|---|
| Image $\mathbf{x}$ | - | - | $128 \times 128 \times 3$ |
| Conv3$\times$3 | - | - | $128 \times 128 \times ch$ |
| ResBlk | - | FRN | $128 \times 128 \times ch$ |
| ResBlk | AvgPool | FRN | $64 \times 64 \times 2ch$ |
| ResBlk | - | FRN | $64 \times 64 \times 2ch$ |
| ResBlk | AvgPool | FRN | $32 \times 32 \times 4ch$ |
| ResBlk | - | FRN | $32 \times 32 \times 4ch$ |
| ResBlk | AvgPool | FRN | $16 \times 16 \times 8ch$ |
| ResBlk | - | FRN | $16 \times 16 \times 8ch$ |
| ResBlk | AvgPool | FRN | $8 \times 8 \times 16ch$ |
| ResBlk | - | FRN | $8 \times 8 \times 16ch$ |
| ResBlk | AvgPool | FRN | $4 \times 4 \times 16ch$ |
| LReLU | - | - | $4 \times 4 \times 16ch$ |
| Conv4$\times$4 | - | - | $1 \times 1 \times 16ch$ |
| LReLU | - | - | $1 \times 1 \times 16ch$ |
| Conv1$\times$1 | - | - | $\hat{K}$ |

Table 7: Discriminator architecture. "ch" and $\hat{K}$ represent the channel multiplier that is set to 64 and the number of clusters, respectively. FRN indicates filter response normalization (Singh & Krishnan, 2020).

# E  T-SNE VISUALIZATION & CLUSTER EXAMPLE IMAGES

## E.1  AFHQ CAT

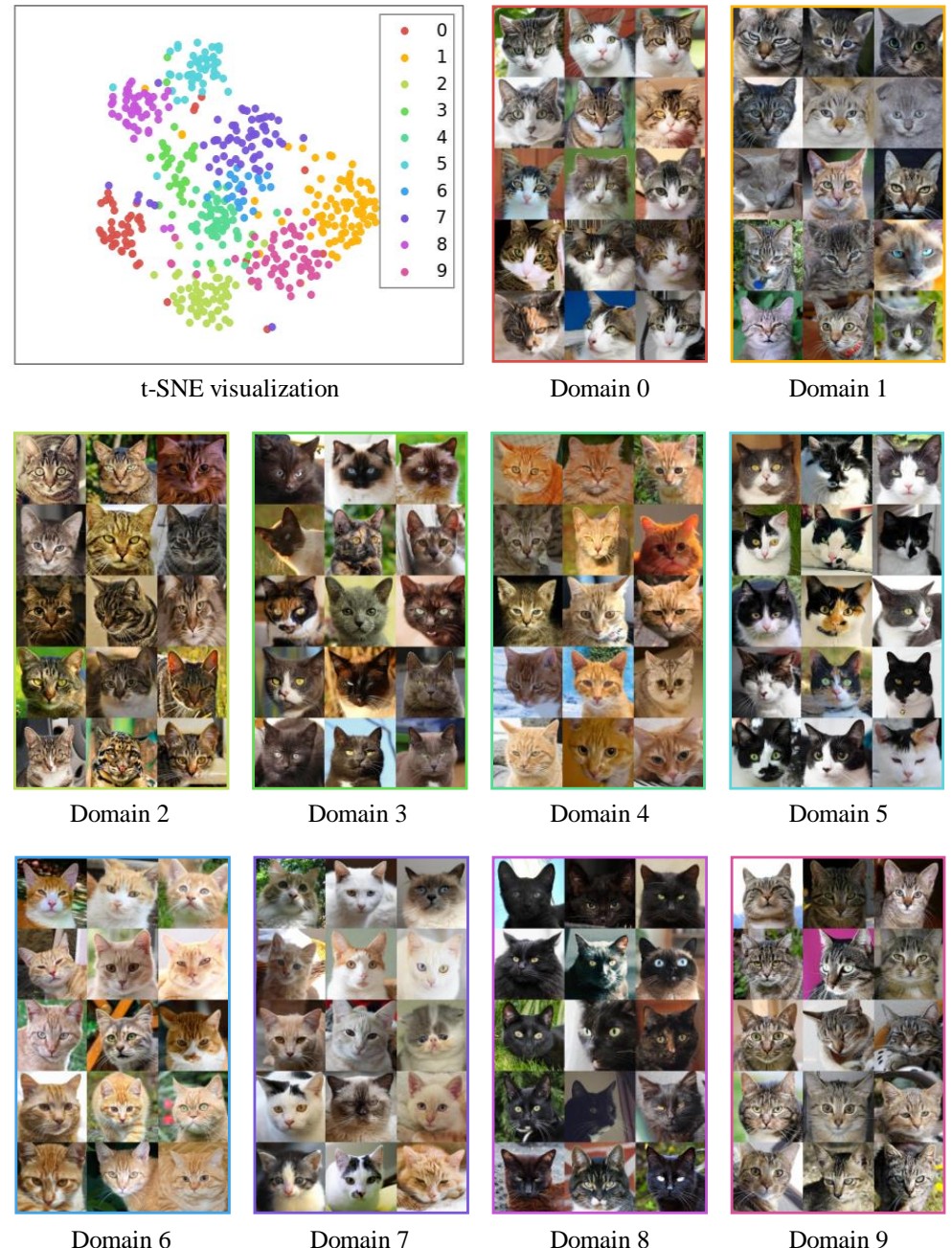

Figure 7: **t-SNE visualization and representative images of each domain.**

E.2    AFHQ DOG

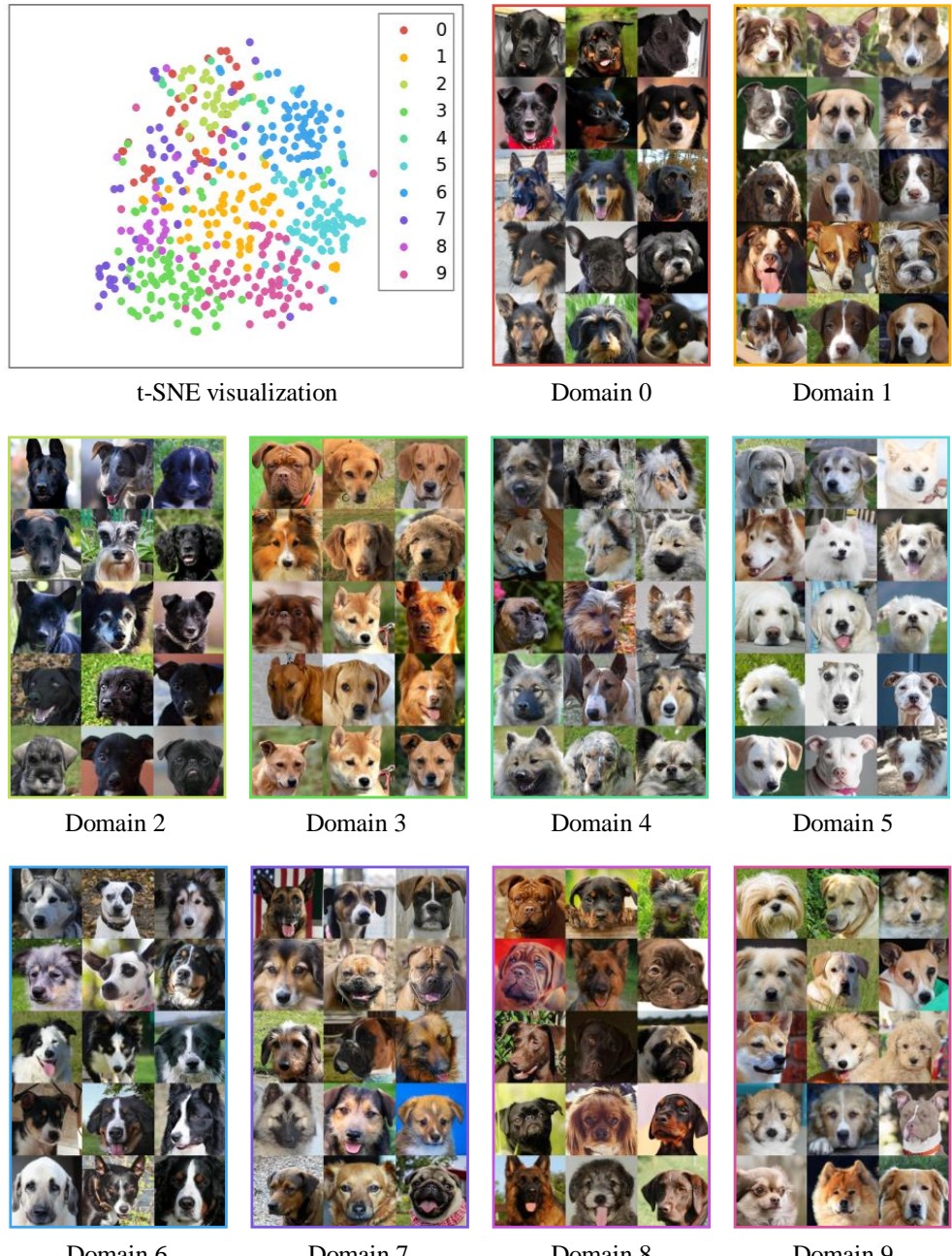

Figure 8: **t-SNE visualization and representative images of each domain.**

E.3    FFHQ

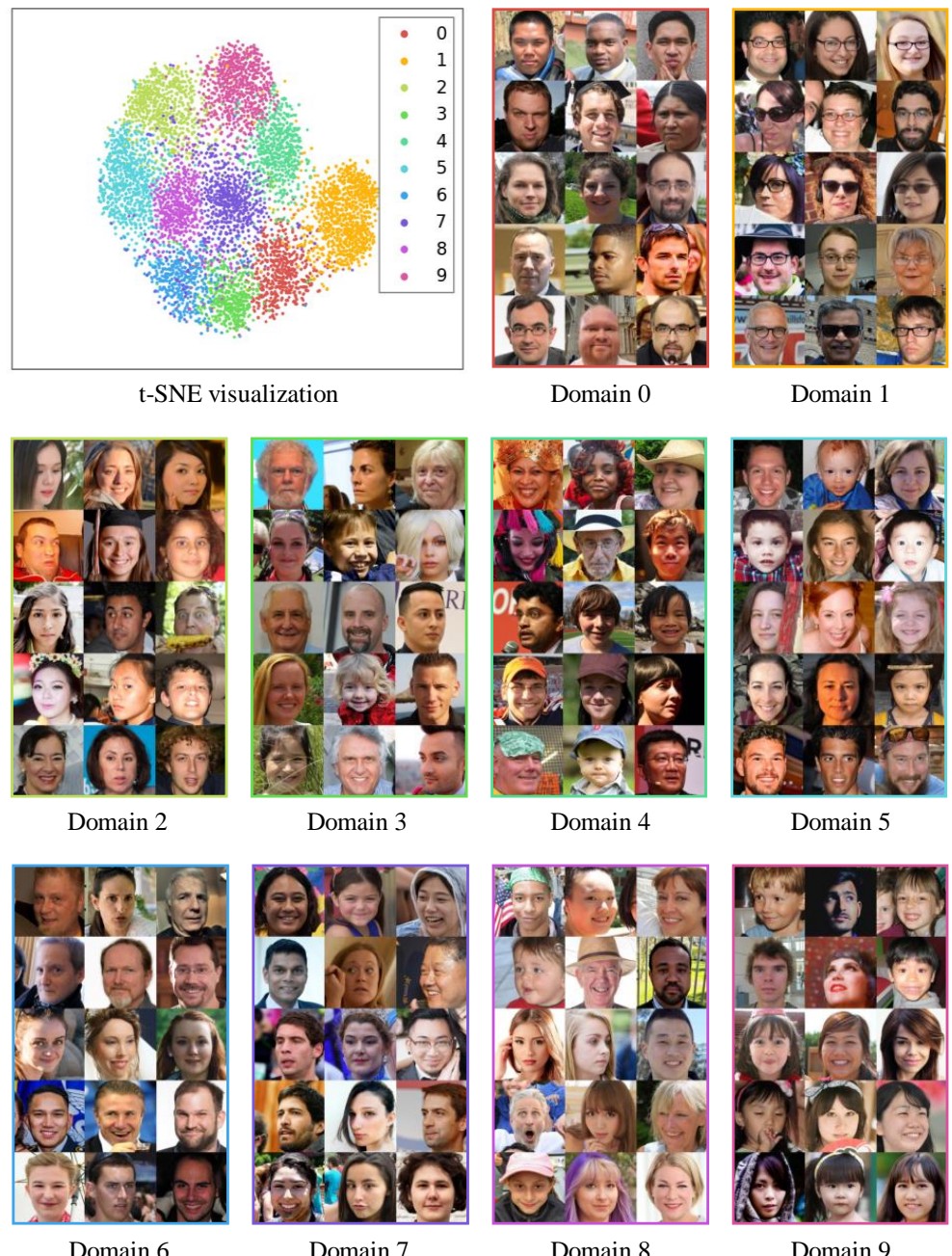

Figure 9: **t-SNE visualization and representative images of each domain.**

### E.4 LSUN CAR

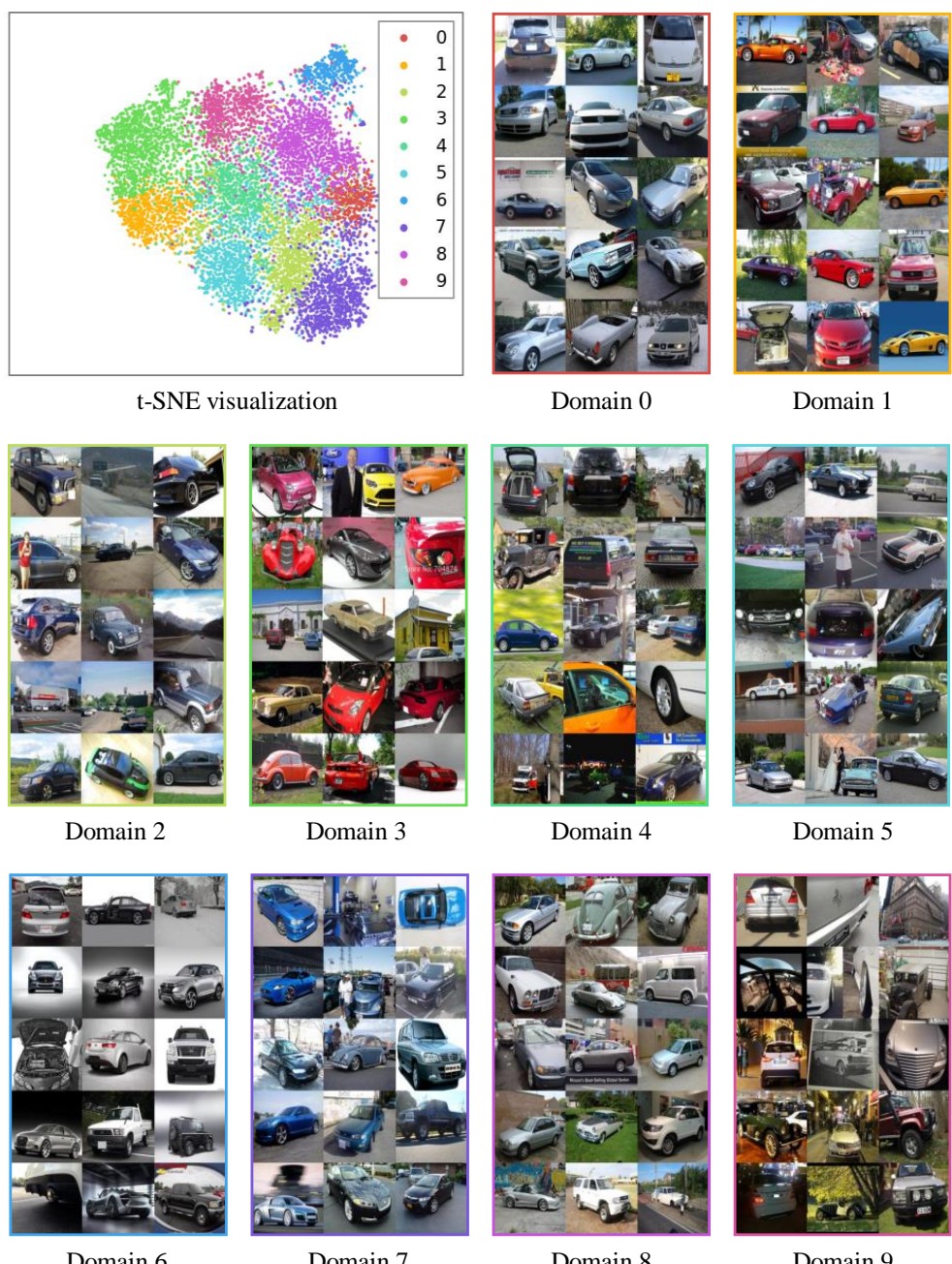

Figure 10: **t-SNE visualization and representative images of each domain.**

## F  ADDITIONAL COMPARISON WITH FUNIT: AFHQ, LSUN CAR AND FFHQ

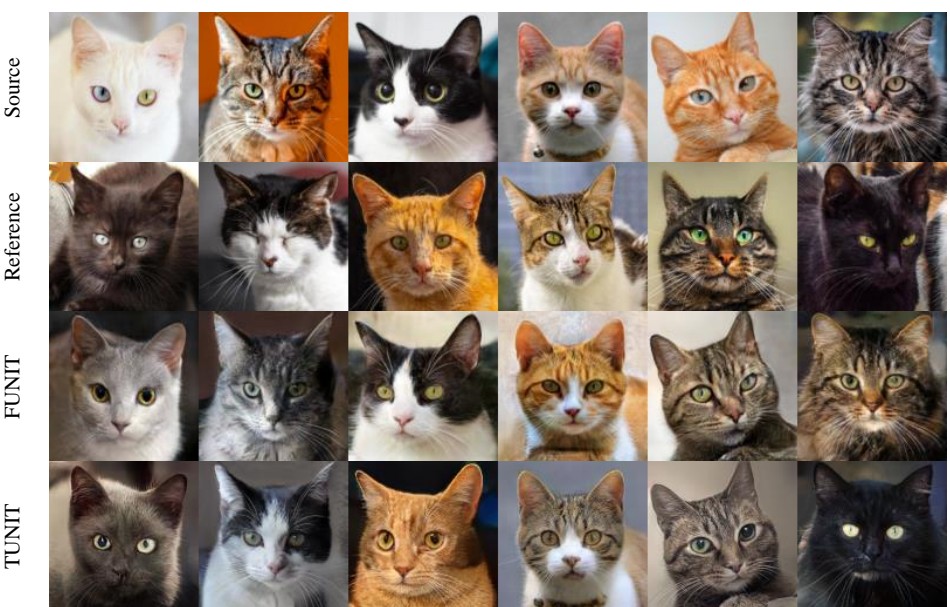

Figure 11: AFHQ Cat, unsupervised reference-guided image-to-image translation results of FUNIT and TUNIT. The content and the style are from the source and the reference, respectively. While FUNIT usually fails to reflect the style of the reference image, TUNIT generates the fake images with the style – color, fur texture.

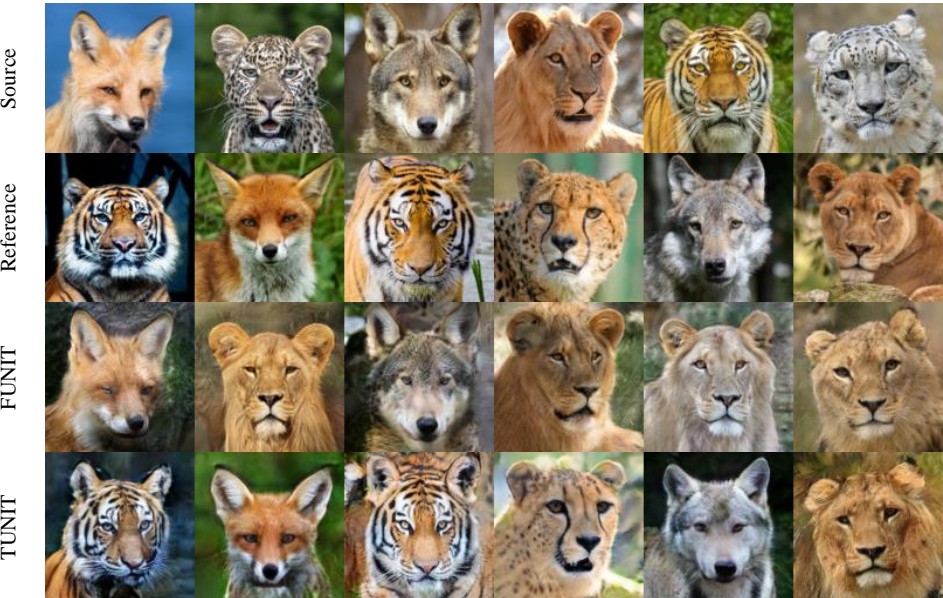

Figure 12: AFHQ Wild, unsupervised reference-guided image-to-image translation results of FUNIT and TUNIT. FUNIT rarely reflects the correct style of the reference image – the species, on the other hand, TUNIT translates the source image to the correct species.

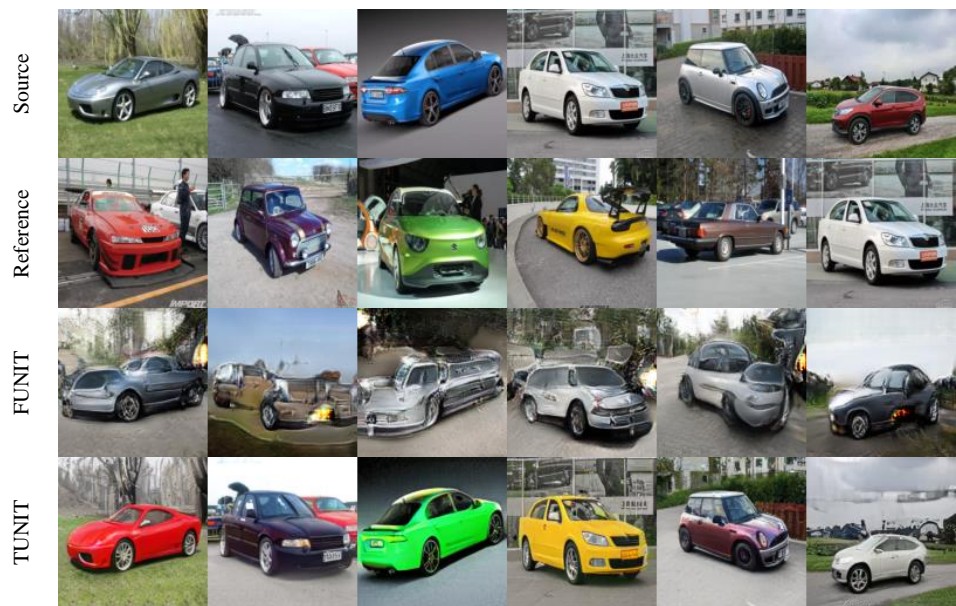

Figure 13: LSUN Car, unsupervised reference-guided image-to-image translation results of FUNIT and TUNIT. While TUNIT generates plausible and changes the color of the source image to that of the reference image, FUNIT not also generates unrealistic image but also fails to changes the color.

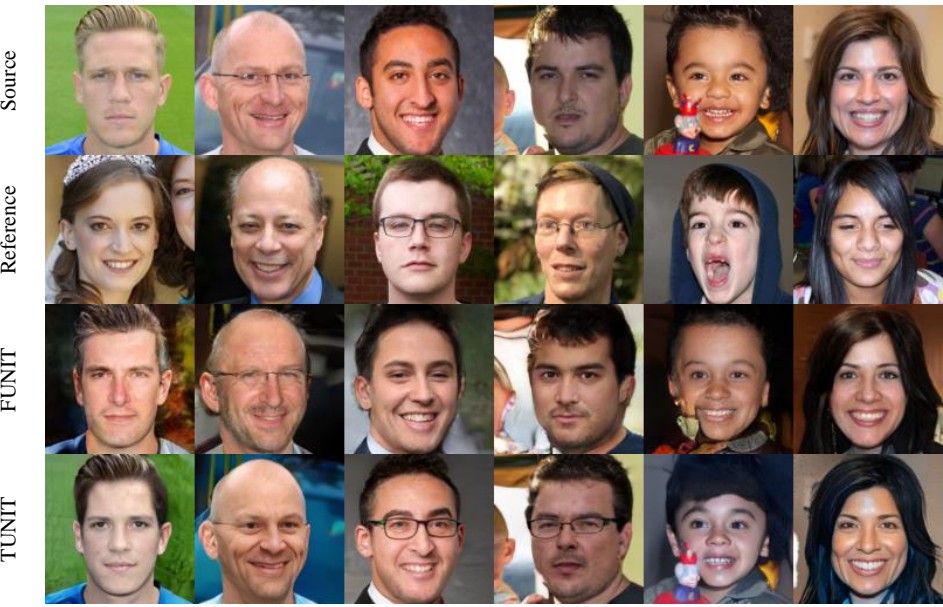

Figure 14: FFHQ, unsupervised reference-guided image-to-image translation results of FUNIT and TUNIT. Our model, TUNIT can remove or add the glasses to the source while preserving the identity better than FUNIT. In addition, TUNIT can change the hair color (last column) and the hair style – especially, bang (fifth column). It is hard to specify the definition of domains in the results of FUNIT while domains of TUNIT are more interpretable.

# G ADDITIONAL RESULTS OF TUNIT: AFHQ, LSUN CAR, FFHQ, ANIMALFACES-10, AND S2W

## G.1 ANIMALFACES-10

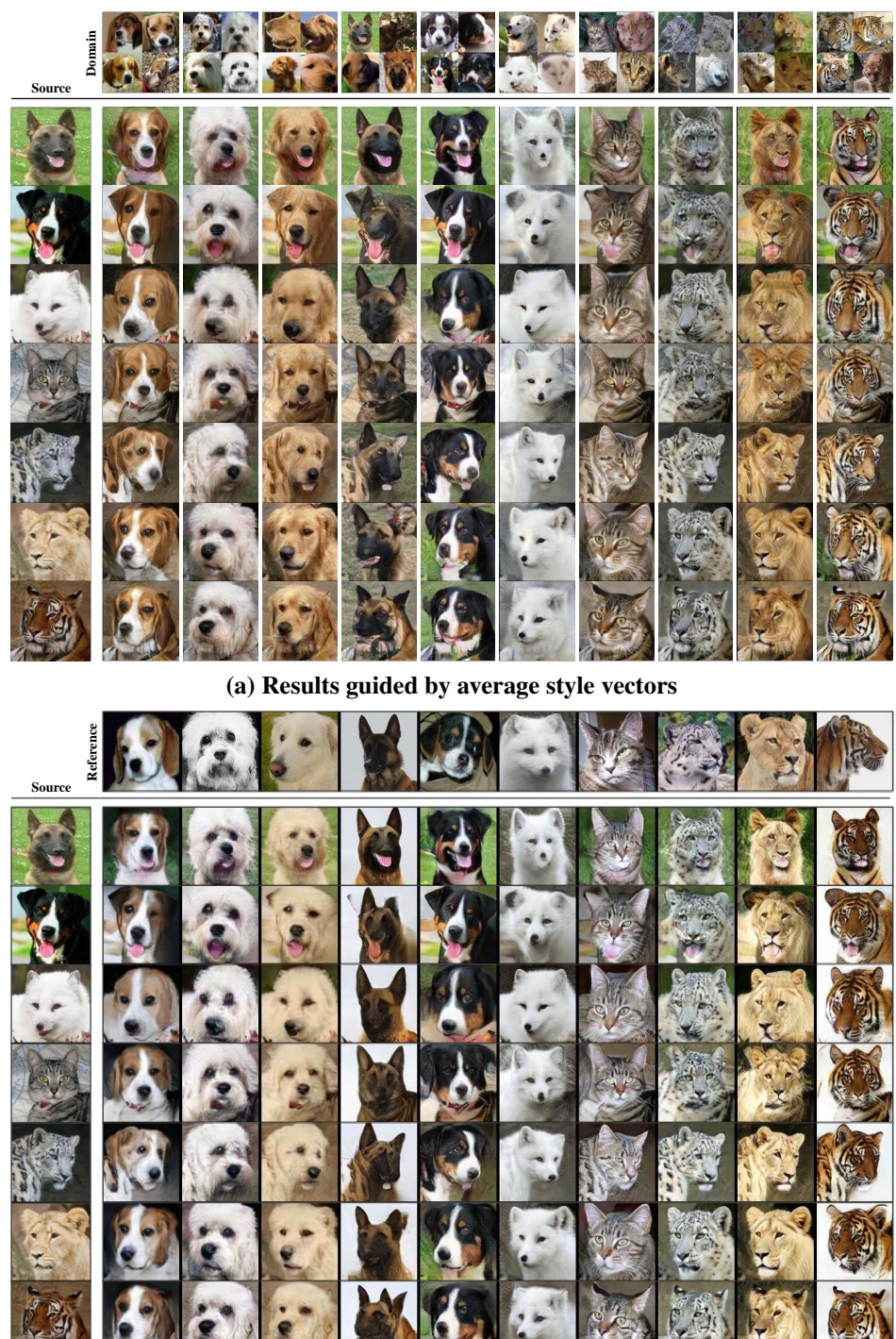

(a) Results guided by average style vectors

(b) Results guided by reference images

Figure 15: AnimalFaces-10, unsupervised image-to-image translation results.

## G.2 AFHQ CAT

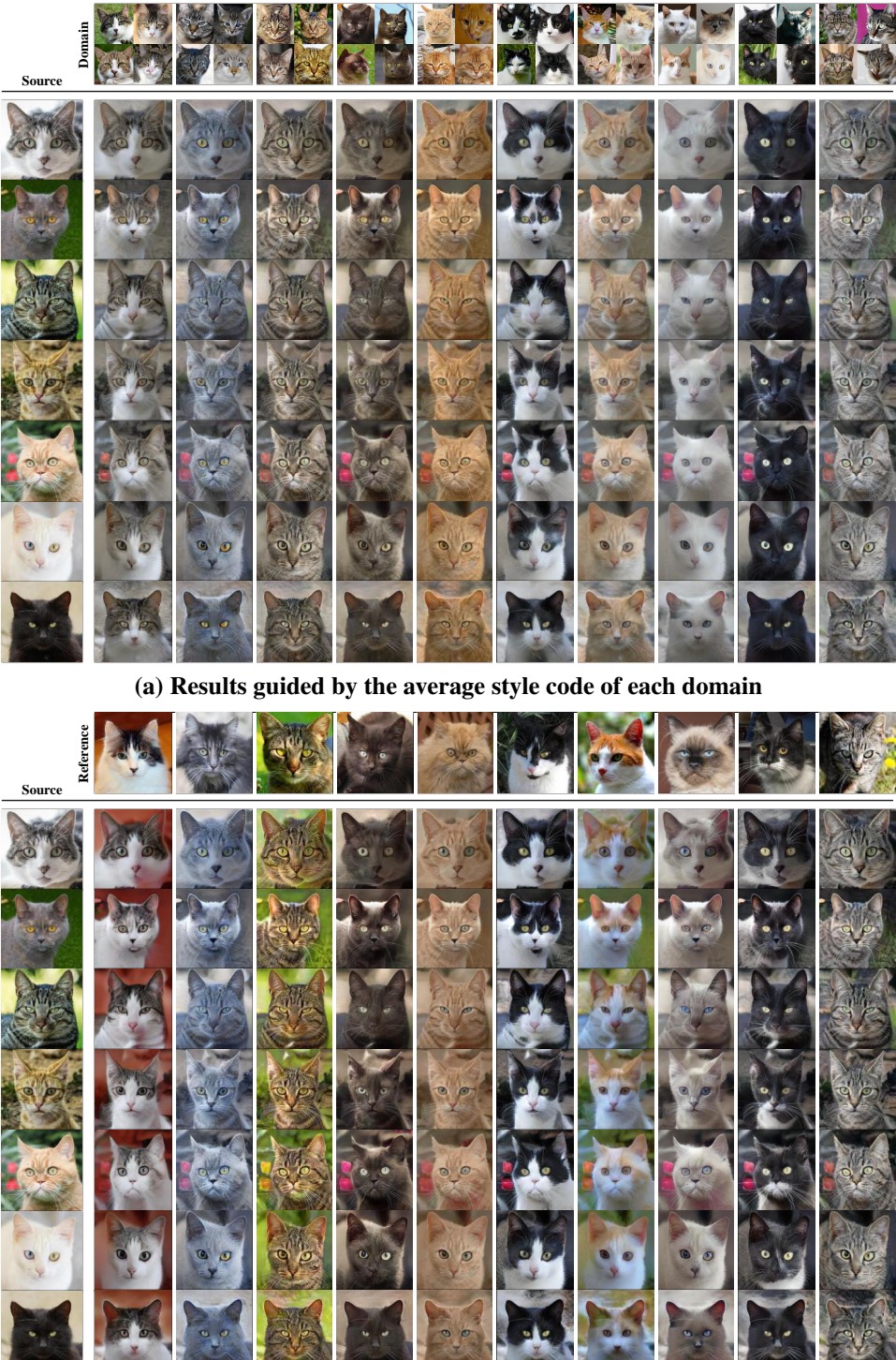

(a) Results guided by the average style code of each domain

(b) Results guided by reference images

Figure 16: AFHQ Cat, unsupervised image-to-image translation results.

## G.3 AFHQ Dog

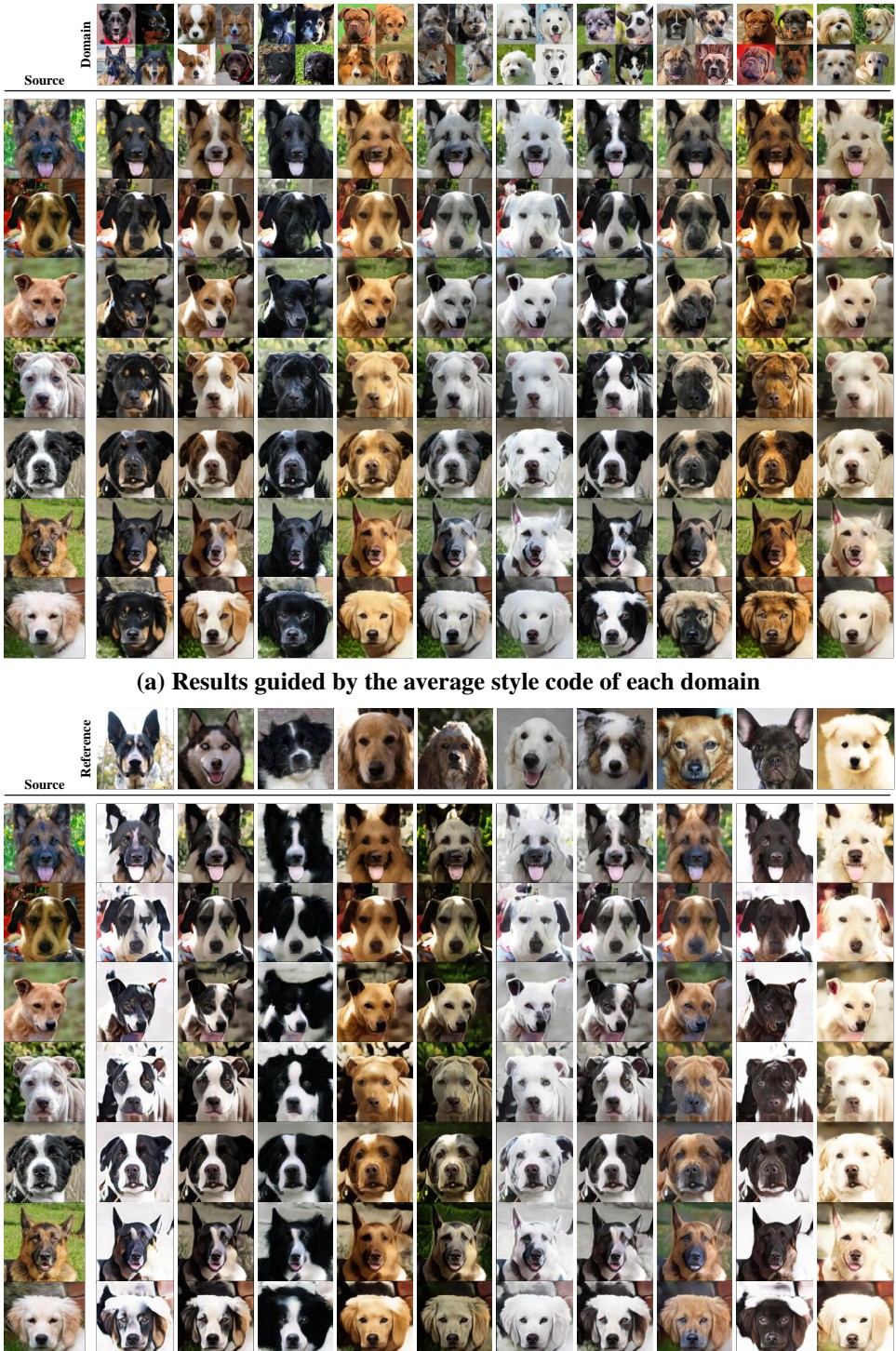

(a) Results guided by the average style code of each domain

(b) Results guided by reference images

Figure 17: AFHQ Dogs, unsupervised image-to-image translation results.

## G.4 AFHQ Wild

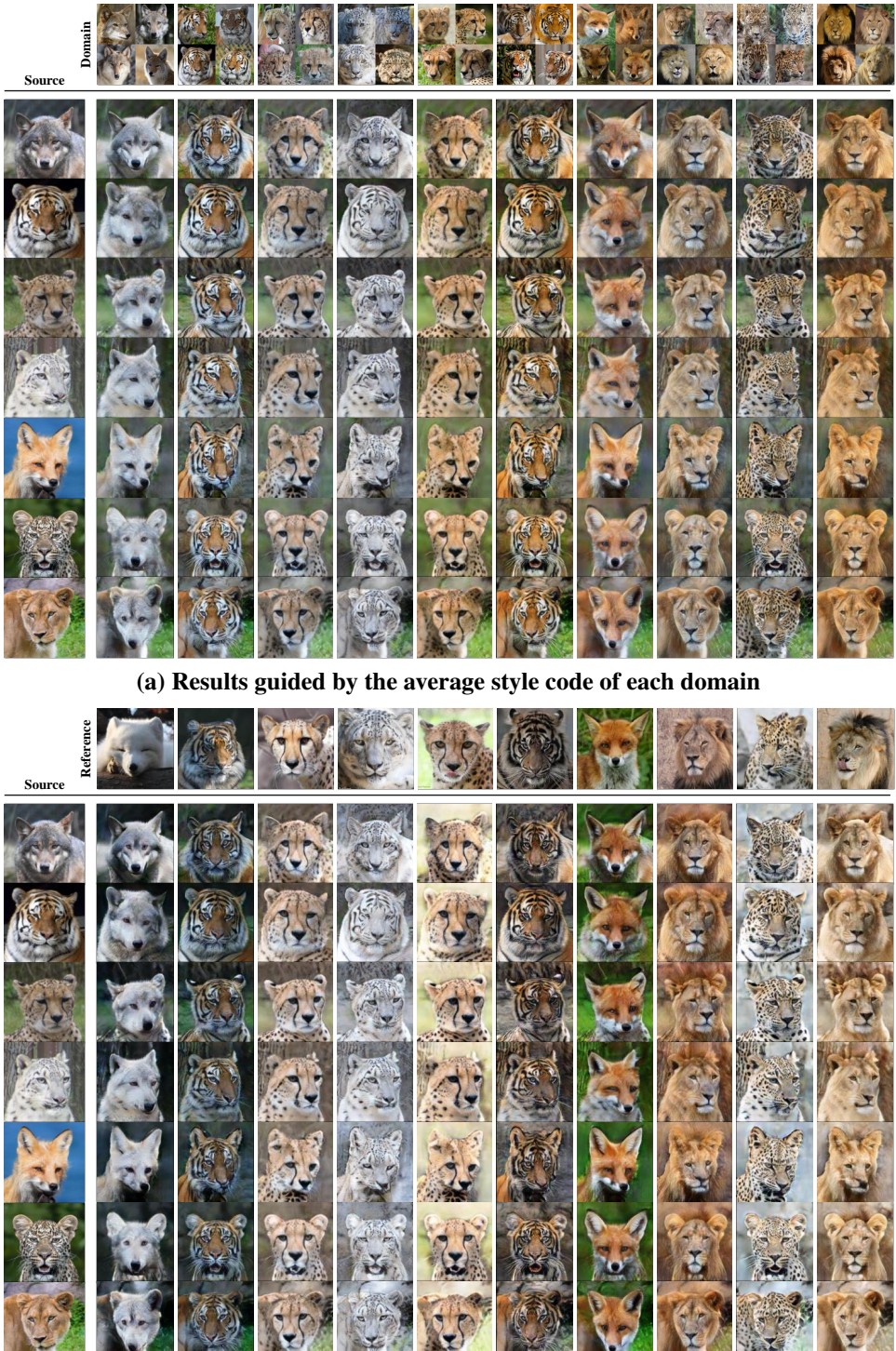

(a) Results guided by the average style code of each domain

(b) Results guided by reference images

Figure 18: AFHQ Wild, unsupervised image-to-image translation results.

## G.5    FFHQ

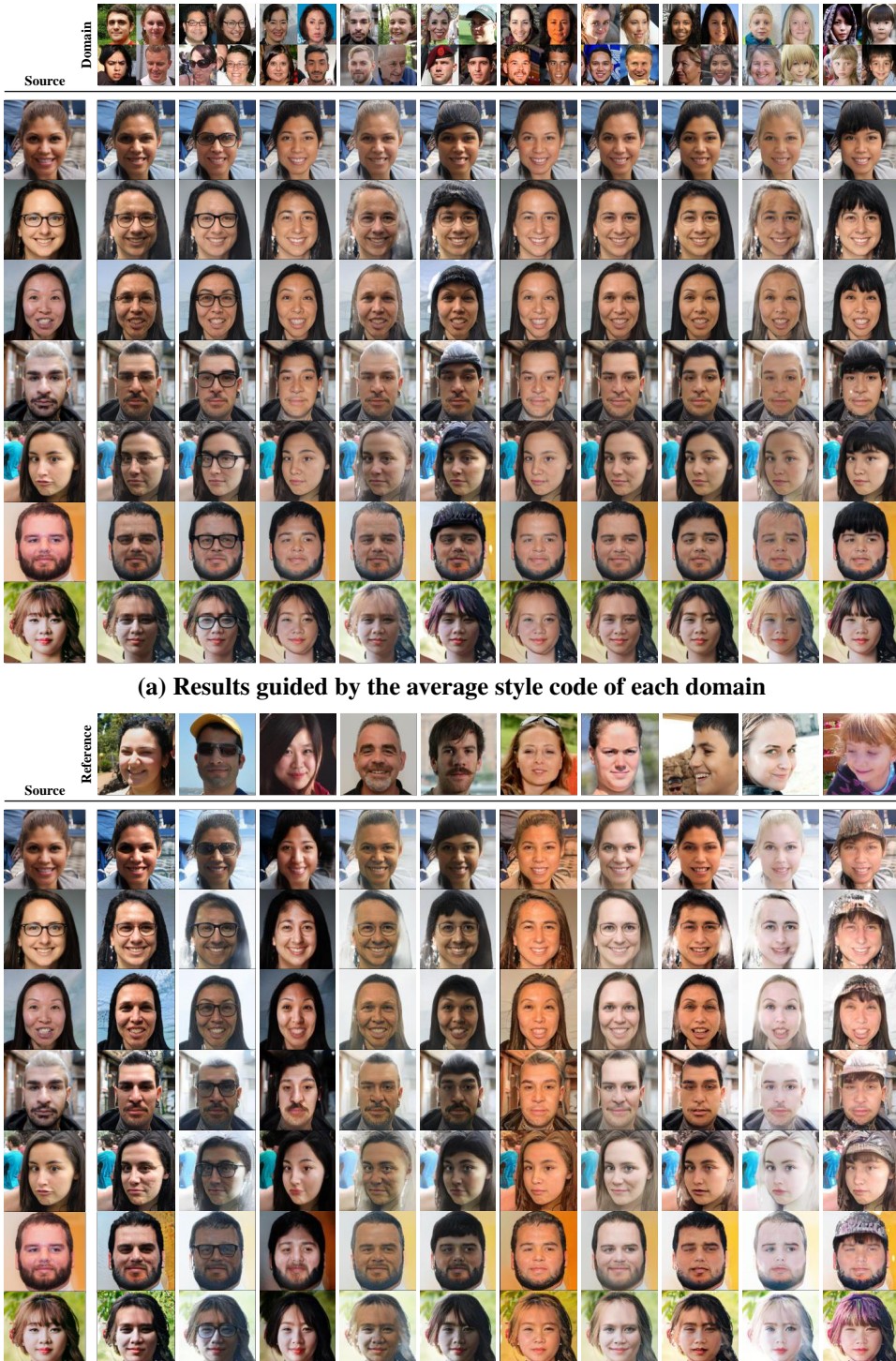

(a) Results guided by the average style code of each domain

(b) Results guided by reference images

Figure 19: FFHQ, unsupervised image-to-image translation results.

### G.6    LSUN CAR

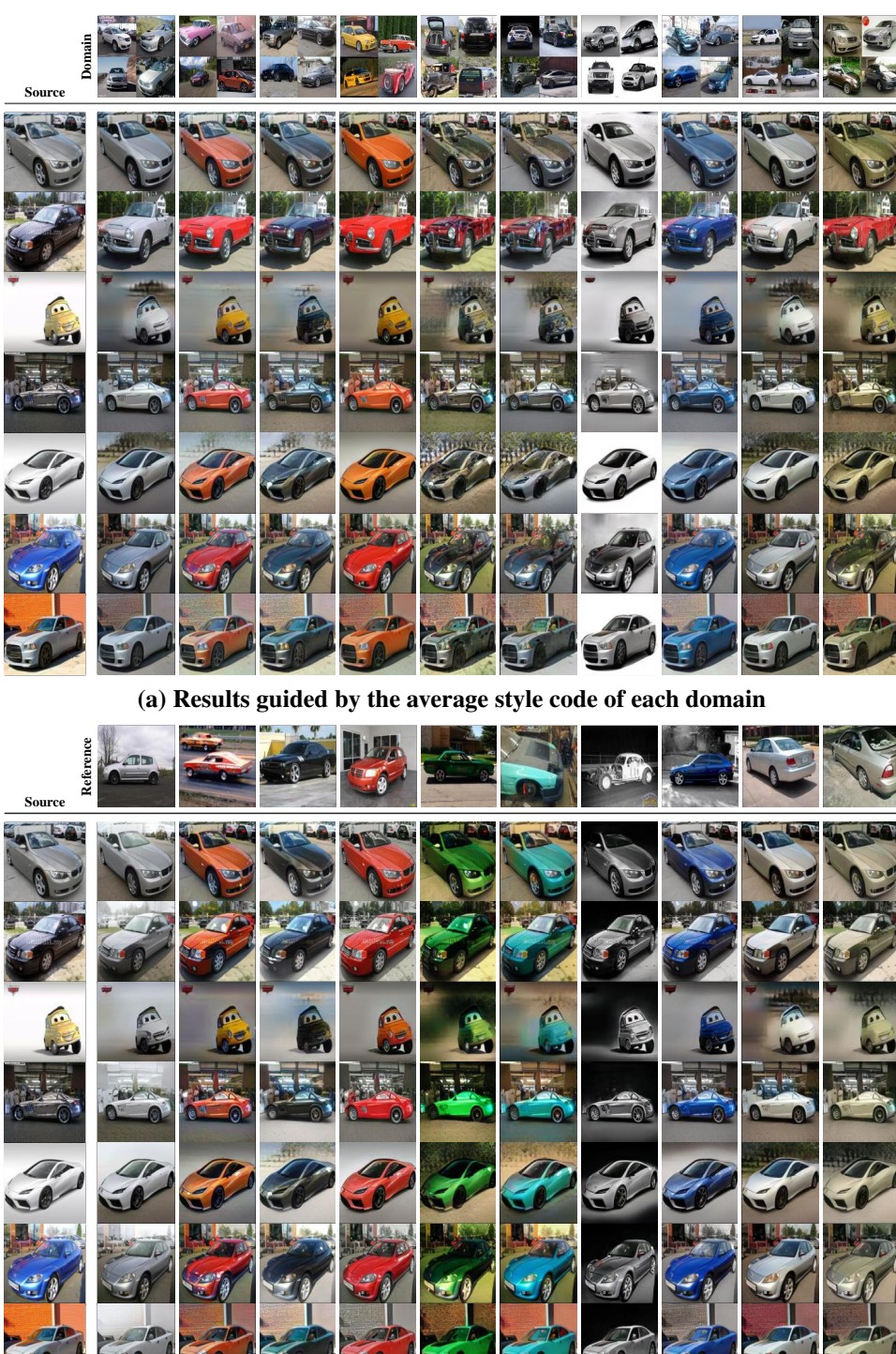

(a) **Results guided by the average style code of each domain**

(b) **Results guided by reference images**

Figure 20: LSUN Car, unsupervised image-to-image translation results.

### G.7    SUMMER2WINTER (S2W)

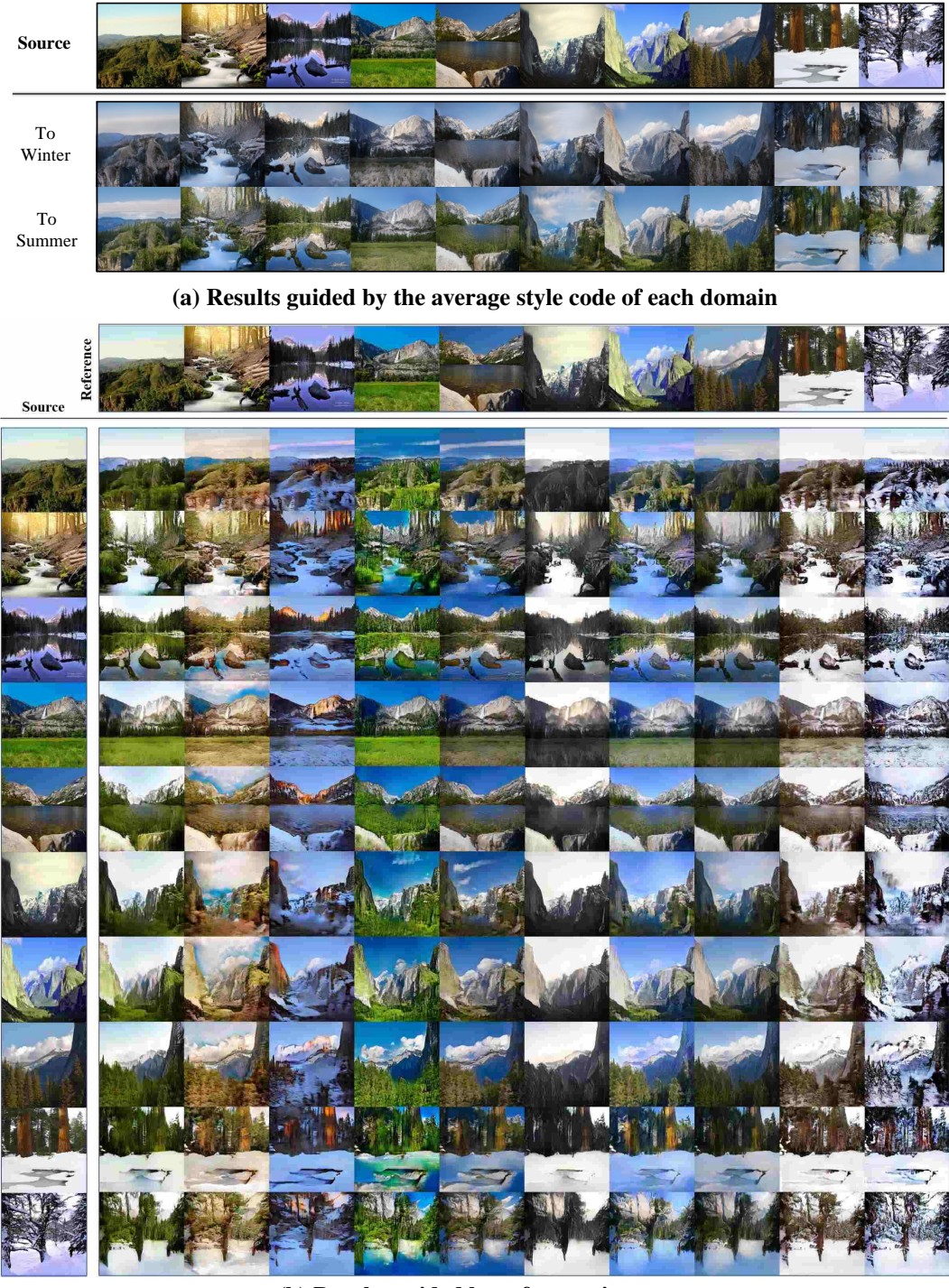

(a) Results guided by the average style code of each domain

(b) Results guided by reference images

Figure 21: Summer2Winter (S2W), unsupervised image-to-image translation results.

## H DIFFERENCE BETWEEN EQUATION (2) AND EQUATION (4)

Equation (2) and (4) have similar forms – contrastive loss, but they are used for different purposes. We use equation (2) to improve the representation power of the guiding network, which affects the performance of the generator and the discriminator. On the other hand, equation (4) is used to enforce the generator to reflect the style of a reference image when translating a source image. To examine the effect of each loss, we train models without either equation (2) or (4) on AnimalFaces-10. The mFID score without equation (2) or (4) is 86.8 and 93.3, respectively. Both models are significantly worse than the original setting (47.7). It means that both equation (2) and (4) should be considered during training. In addition to the purpose, they are different in terms of the way to choose positive pairs. We use a real image and its randomly augmented version as a positive pair in equation (2) while we use the translated image and reference image as a positive pair. In summary, the role of equation (2) is to enhance the representation power of the guiding network and lead the guiding network to learn how to encode the style vector in terms of a style encoder while the role of equation (4) is to guide the generator to learn how to interpret the provided style vector as a form of the output image.

## I FID AND LPIPS ON UNLABELED DATASET

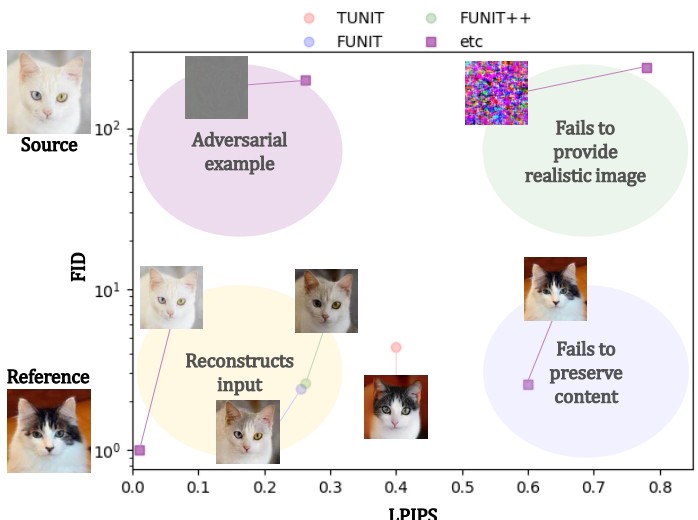

Figure 22: **LPIPS and FID of models and their status.**

We also utilize LPIPS to evaluate the models in addition to FID and D&C. However, LPIPS is not proper to evaluate the loyalty for reflecting the reference image and the fidelity of images, we use LPIPS with FID. Figure 22 shows the result. It is clear that a model with high FID and LPIPS generates a noise-like image. Even if FID is low, a model with high LPIPS also fails to conduct the reference-guided image translation, because it does not preserve the structure of the source image. The model with low LPIPS and high FID might be an adversarial example of LPIPS. We generate the image via optimization on LPIPS. If a model exhibits low FID and LPIPS, it might not reflect the visual feature of the reference image enough. The simple combination of LPIPS and FID can detect several failed models but can not evaluate the loyalty for the reference image. We suggest that the rigorous way to combine several metrics for the quantitative evaluation of the reference-guided translation might be a interesting future work.

