# OpenReview forum: "Rethinking the Truly Unsupervised Image-to-Image Translation"
_ICLR.cc/2021/Conference — Reject_

### Official Review · AnonReviewer4 · 2020-10-25

**Rating:** 6
**Confidence:** 4

**Review:**

This paper tackles the problem of unsupervised image to image translation without making any assumptions about the existence of input pairs or input sets. To solve this task, the authors utilize a guiding network that generates a pseudo label (domain label in this case) as well as a style code which are then provided to a generator network. The generator given a source image produces an output that follows the structure of the source yet preserves the style and "domain" details of the reference image. These  networks are trained jointly with adversarial, contrastive and reconstruction losses. Results are reported on labeled and unlabeled datasets

Pros:
- What's quite neat is the incorporation of SSL to handle the lack of labels where augmentation is applied to the reference image with the goal of maximizing the mutual information between the output distribution of the input and the augmented image.
- Very well documented results, Table 1 and Figure 3 are well designed to show how each of the components of this method (or alternatives that the authors could have used) affect the unsupervised translation results.
- Results-wise what is provided is actually quite compelling, it's easy to understand the impact of style from the source and the "structure" of the reference images on the generated results at least for the animals and the cars.

Cons:

- I am somewhat skeptical of the statement "our model is the first to succeed in this task in an end-to-end manner" as a selling point of this paper. For example, a prior work that comes to mind is CUT [1] where the authors also target the unsupervised image-to-image translation and also utilize contrastive losses in patches of images. One could argue that in CUT they still require a set of domains whereas this paper does not but one could first do some clustering and then apply CUT which is quite similar to what this paper is proposing. I think it would be fair to ask for a discussion/comparison with that paper.
- While the idea of estimating a pseudo-label from the encoder in a SSL fashion is well-designed the rest of the architecture is similar to other recent works that perform "exemplar-based image synthesis". In [2] for example the authors feed a reference image that serves as an exemplar from which they extract class-specific styles which are then fed to the generator that synthesizes the output image. This procedure is similar to what's happening with the style code that is fed to the generator in this paper.
- Is joint training better compared to sequential for the guiding and generator networks? The results from Table 1 indicate that if one looks at mFID or Acc this could be dataset-specific. I would encourage the authors to potentially elaborate a little more on that since what's currently stated is that "we confirm that the joint training of style encoder and clustering improves overall performances"

Minor Comment:
- From what I understand from Figure 2 the estimated domain label from the encoder is fed to the discriminator but there is then a "detachment" so as we do not backprop through the encoder. I would encourage the authors to potentially expand this a little in the paper as it seems to be that besides noting it in that Figure there's no further description of this in the paper.

Overall:
This is an interesting paper, very well written, and easy to follow targeting a hard and under-researched problem. The guiding network is nicely designed and the results seem quite  strong in terms of texture transfer while maintaining the source shape although fidelity-wise the still have notable artifacts (easier to observe in faces). My main concerns are more related to how some statements related to novelty or training findings (joint vs sequential) are presented and how much new information does the reader get from this paper.

References
[1] Contrastive Learning for Unpaired Image-to-Image Translation ECCV 2020
[2] SEAN: Image Synthesis with Semantic Region-Adaptive Normalization CVPR 2020

Update:
I'd like to thank the authors for their detailed response to my comments.

- I respect the argument that CUT requires a model per pair of domains hence it does not scale to 10 domains. However  presenting the problem in such an aspect ("we evaluate only on multi-domain datasets such as the AnimalFaces, Food") limits a little the prior work that one can evaluate against. In my humble opinion the authors constrain the definition of unsupervised image-to-image  translation such that their proposed approach "is the first to succeed in this task" whereas I also think CUT is an unsupervised image-to-image translation method that "succeeds in this task". If the  argument was presented in the multi-domain setup I would be willing to buy it but as it is presented currently I find the first contribution as an overstatement.

- As for SEAN my point was not to compare against it but that the architectural design of this work is not far from what prior work is already doing.

- Sounds good thanks for the explanation.

Having said that I still think this is a solid submission with interesting findings in an under-researched problem hence I do recommend acceptance.

---

> ### Author Response · Authors · 2020-11-16
> **Official response to Reviewer #4**
>
> We appreciate the reviewer4 for taking the time and effort to review our paper and for the helpful comment. We are good to hear that our contribution is recognized as neat and the experimental results are well documented. Here we address the three main concerns:
>
> **W1) Relationship between TUNIT and CUT**
>
> We would like to provide our reasons for why CUT is not a proper baseline for TUNIT.
> First, CUT conducts the unpaired cross-domain image-to-image translation that requires 90 models to conduct the ten-domain translation task, therefore, CUT is not a proper baseline.
> Second, the purpose of the contrastive loss is different. CUT utilizes the contrastive loss to preserve the content of the source image and to remove the necessity of the cyclic loss and additional networks. On the other hand, we adopt the contrastive learning scheme to enhance the representation of the guiding network and to enforce the generator to reflect the style of the reference image. We agree that CUT can be cited as a related work in terms of using contrastive learning in the translation task, however, CUT is not adequate to be a baseline of TUNIT.
>
> **W2) Relationship between TUNIT and SEAN**
>
> We would like to emphasize that SEAN and TUNIT are different methods that target different tasks using different levels of supervision. SEAN is a paired cross-domain image-to-image translation model whose goal is semantic image synthesis. Thus, SEAN is more similar to SPADE that synthesizes an RGB image from a semantic segmentation map or example-guided style-consistent image synthesis [1]. To perform image synthesis, unlike TUNIT, SEAN requires full image-level supervision---the ground-truth pair of an image and a semantic segmentation map. The only intersection between TUNIT and SEAN is that they both use style vectors. However,  it is a commonly used practice in reference guided image-to-image translation. In addition, SEAN aims to synthesize an RGB image having a different style for each class of a semantic segmentation map, while TUNIT conducts the translation of an RGB image to another RGB image having the style of a reference image.
>
> **W3) "Is joint training better compared to sequential training?"**
>
> The difference between F and G in terms of clustering accuracy is around 1% in Table 1. Because our main goal is to conduct the translation well even without set-level supervision, we focus on the translation performance. In terms of the translation, model G outperforms except for mFID on AnimalFaces-10. While the difference in mFID on AnimalFaces-10 is 1.7 (G-F), the difference on Food-10 is -8.9(G-F). It means that G is more stable in terms of inter-dataset performance. And practically, the end-to-end training scheme is more convenient. To sum up, because of the stability of the performance in terms of the translation task and convenience, we currently state “we confirm that the joint training of style encoder and clustering improves overall performances".
> We change the statement to “By comparing model F and G, we confirm that they are comparable in terms of clustering and the model G is more stable in terms of inter-dataset performance. Therefore, we adopt the joint training of style encoder and clustering as our final model.”
>
> This is reference:
>
> [1] Wang, Miao, et al. "Example-guided style-consistent image synthesis from semantic labeling." Proceedings of the IEEE Conference on Computer Vision and Pattern Recognition. 2019.

---

> ### Author Response · Authors · 2020-11-25
> **Thanks for reply and support!**
>
> Thanks for the reply and explanation!
>
> We understand your points about CUT and SEAN and we would like to provide the reason for the past claims.
>
> **CUT**
>
> We understand that our claim of "the first to succeed in this task in an end-to-end manner." seems to be an overstatement and we respect your argument that CUT might be the work "succeeds in this task".
>
> We would like to provide why we claimed that TUNIT is the first work that solves this problem.
>
> - The works published in ECCV are considered contemporaneous and CUT is published in ECCV (The deadline of ICLR 2021 is 02 Oct. and, following the review guideline of ICLR'21, the papers published after 02 Aug. are considered contemporaneous -- the conference day of ECCV is 23-28 Aug.). Related to the timestamp, we can provide another reason, however, it might violate the double-blind so that we do not provide it here.
>
> - In this version, we compare and evaluate TUNIT in multi-domain settings, and it limits the range of the prior works. Even if it is not in the main text, we show that our work can be applied to the cross-domain translation by providing the qualitative results for the summer2winter dataset. We will add the quantitative result and comparative evaluation with MSGAN to the appendix section for future readers (It is not a strong claim that TUNIT outperforms any other cross-domain works but is a mild demonstration that TUNIT also works on the cross-domain dataset.).
>
> We will clarify the sentence not to be an overstatement.
>
> **SEAN**
>
> We understand the point. Thanks for the clarification.
>
> - The overall architectural design to adopt a style encoder, content encoder and decoder might be similar to SEAN and TUNIT. However, this type of design is widely adopted in image-to-image translation work. For example, MUNIT and FUNIT also adopt two types of encoders and one decoder. We want to emphasize that our TUNIT utilizes the style encoder that also works as a domain clusterer. By doing so, TUNIT can leverage the representation learned during clustering for the style encoding. This is a different point in terms of architectural design.
>
> Again, we appreciate the valuable comment to improve our work!

---

### Official Review · AnonReviewer1 · 2020-10-26
**Official Blind Review #1**

**Rating:** 6
**Confidence:** 5

**Review:**

The paper introduced a more challenging setup under the problem of unpaired image-to-image translation, where no domain labels or image sets are provided. Towards solving this problem, the paper proposed a TUNIT framework with a guiding network, which could encode style codes for the generator and predict the domain labels for the discriminator. The paper is well-written with nice figures. Abundant experiments are conducted and lots of generation results further verify its effectiveness. But there still exists some weaknesses:

1. A contradiction exists that the author claimed the number of domains "K" is an unknown property of the dataset (in the 2nd line of Sec 2), but using the guiding network to predict the domain labels under the assumption that "K" is known (Eq. (1)).

2. The description of "Style contrastive loss" (Eq. (4)) is confusing. The author claimed that the network may ignore the given style code $\tilde{s}$ and synthesize a random image from the same domain $\tilde{y}$ without Eq. (4), however, as we learned from Eq. (2), the "given style code $\tilde{s}$" and the "style codes of images from the same domain $\tilde{y}$" are similar to each other. A contradiction exists. In my point of view, Eq. (4) is used to make sure that the generated images show the same visual style features as the input reference images. More explanations are expected here.

3. Since Eq. (2) and Eq. (4) have similar equations and maybe there are also some overlapping functions. I expect more discussions regarding the relations and differences between these two losses. Furthermore, how does the network perform if removing one of them?

4. The usage of contrastive loss and image queue is borrowed from MoCo [A], so it's better to acknowledge it and mention the relations.

[A] Momentum contrast for unsupervised visual representation learning. CVPR 2020.


Given the above weaknesses, I gave a rating of 5 in the initial comments and I will increase my score if the author could 1) provide better explanations of the contradictions, 2) discuss and add ablation studies on the functions of Eq. (2) and Eq. (4).



===========================================================================================================

Update:

Thanks to the author's response, solving my concerns on this work. I decide to increase my score to 6. Besides, I suggest supplementing the mentioned discussions and empirical comparisons between Eq. (2) and Eq. (4) in the manuscript, as least in the appendix.

---

> ### Author Response · Authors · 2020-11-16
> **Official response to Reviewer #1**
>
> We appreciate the reviewer1 for the helpful and detailed review. We are grateful to hear that our paper is well-written and has extensive experimental results with nice figures. Here we address the four main concerns:
>
> **W1) Contradiction on $K$**
>
> Sorry for the confusion. It is not a contradiction but a misunderstanding due to  our insufficient explanation Indeed, because one never knows the ground truth number of clusters in advance, in equation (1), $K$ should be changed to $\hat{K}$, which is selected by the user before starting the training. To avoid this confusion, from now on, we will denote the estimated number of clusters as $\hat{K}$ and its ground-truth as $K$. For example, we conduct the experiments on Section 3.2 without any information related to the true number of clusters ($K$) so we set an arbitrary number ten; $\hat{K}$=10. In addition, to investigate the sensitiveness to hyperparameter change, we conducted an experiment about the relationship between $\hat{K}$ and the performance in Section 3.3. The actual number of domains in AnimalFaces-10 is ten but we trained TUNIT with $\hat{K}$=1,4,7,10,13,16,20 and reported mFID and D&C. During the rebuttal, we also trained TUNIT with $\hat{K}$=50, 500, 1000, whose mFID scores are 63.8, 67.2, and 66.9 on AnimalFaces-10, respectively. Here, $K$=10. To make the description clear, we change K to $\hat{K}$ and add an explanation about the selection of $\hat{K}$ in Section 2 as “We note that $\hat{K}$ is arbitrarily chosen before the training. Please refer to Section 3.3 for more details.”
>
> **W2) The description of "Style contrastive loss" (equation (4)) is confusing.**
>
> Sorry for the confusion. It seems like our intention is not delivered correctly.  Regarding the description of  equation (4) - “In order to prevent a degenerate case where the generator ignores the given style code $\tilde{s}$ and synthesizes a random image of the domain $\tilde{y}$, we impose a style contrastive loss:”, our original intention is actually the same as what you said.  Here, in “to prevent the generator from ignoring the style code”, our intention was to say that it enforces the generator to reflect the style of the reference image.
> We modify the main text to clarify the meaning. For more explanations about equation (2) and (4), please check the below W3.
>
> **W3) Difference between equation (2) and (4) and their role**
>
> Equation (2) and (4) have similar forms but are used for different purposes. Equation (2) is used to improve the representation power of the guiding network, which affects the performance of the generator and discriminator. On the other hand, equation (4) is used to make the generator reflect the style of a reference image when translating a source image.
> The mFID score on AnimalFaces-10 without either equation (2) or (4) is 86.8 and 93.3, respectively, both of which are much worse than the original setting (mFID of 47.7). This result shows that both equation (2) and (4) should be considered during training.
> An additional difference between equation (2) and (4) is the way to choose positive pairs. As in a typical contrastive learning setup, equation (2) uses a real image and its randomly augmented version as a positive pair. On the other hand, equation (4) uses the translated image and reference image as a positive pair.
> In summary, the role of equation (2) is to enhance the representation power of the guiding network and lead the guiding network to learn how to encode the style vector in terms of a style encoder while the role of equation (4) is to guide the generator to learn how to interpret the provided style vector as a form of the output image.
>
> **W4) Acknowledgement of MoCo**
>
> In the original manuscript, we have cited MoCo in the related work section (see Appendix) and in the method section (please see the description of equation (2)). However, we also admit that it would make the relation much clearer if we acknowledge it in equation (4) as well. Reflecting the reviewer's opinion, we add the citation and description in the main text.

---

> ### Author Response · Authors · 2020-11-20
> **Thanks for the reply and the decision!**
>
> Thanks for the reply and the decision. We add the discussion about Eq. (2) and Eq. (4) to the appendix (Appendix H) and we also add a sentence pointing to the discussion at Eq. (4).
>
> Again, we sincerely appreciate the reviewer's valuable comment to improve our work.

---

### Official Review · AnonReviewer2 · 2020-10-27
**Fully unsupervised I2I with extensive experimental results, some concerns about experiments**

**Rating:** 6
**Confidence:** 5

**Review:**

This paper proposes an unsupervised I2I translation method TUNIT where there is no any supervision signal. Built upon the existing FUNIT work which disentangle style and content, the proposed method additionally adds a pseudo label prediction branch to separate domains based on maximizing the mutual information. Experimental results look pretty solid. Some of my concerns are listed below:

(1) Without any supervision, the selection of K sounds very important. I'm interested in a more crazy value for K in Table 2. If a collection of images are given and the estimation of K is unlikely to happen, how should K be selected?

(2) The experiment is based on randomly selecting 10 classes from the dataset. Not sure how much difference or similarity exist between those classes. It is better to show the average performance of several 10-class sets.

(3) I like the baseline comparison with K-means but it sounds like a very old clustering method. Will more advanced clustering method bring further improvement on performance?

(4) I suggest adding another line in Table 1 which is about TUNIT + using domain labels (with a classifier to predict real labels). This could serve as a bar on performance to let readers be clear about the gap between w/ and w/o labels.

(5) I feel this work mostly focuses on unpaired data case. To my understanding, the claim "better than supervised models" should be about "better than set-level supervised models". If paired data is given, the proposed method might not beat its performance.

=======

Update:

Thanks for the feedback from authors. Mostly my concerns are addressed. I suggest adding the discussion on the selection of K in the draft as this is important for readers to know when facing just a collection of images.

---

> ### Author Response · Authors · 2020-11-16
> **Official response to Reviewer #2**
>
> We appreciate the reviewer2 for valuable and detailed comments and suggestions. We are glad to hear that you acknowledged our contribution and experiments. Here we address the five main concerns:
>
> **W1) How to select $K$ without any information**
>
> This is a very sharp point. In short, practitioners can safely set $K$ as a reasonably large number (up to 100 times of the actual number of domains), or (if one really wants to find the best performance) study different $K$’s in log scale (e.g., first try $K$=1, 10, 100 and 1,000 and narrow down the search space.) The reason is as follows.
> As shown in Table 2 and Figure 6, our model is quite robust against the variation of $K$ unless it is too far from the true number of domains (e.g., $K$=1 when there are actually ten domains). Based on this, we suggest setting $K$ as the value from 70%($K$=7) to 200%($K$=20) of the expected (or unknown true) number of domains. Yet, it would be nice to have more buffers since we never know the true number of domains in advance. To address this, we conducted additional experiments to see the consequence of setting very large $K$ (10,000%) than the actual number of domains (ten). We train the models with extreme $K$’s -- 50, 500, and 1,000. When we set $K$ to 1,000, each cluster contains roughly ten images. The mFID of the models with $K$=50, 500, 1,000 is 63.8, 67.2, and 66.9 on AnimalFaces-10, respectively. It shows that increasing $K$ only slightly degrades the performance yet remaining in a reasonable range. Here, we confirm that TUNIT performs reasonably well even with an extremely large $K$. We provide the t-SNE visualization of $K$=50, 500, 1000 in the general comment.
>
> === EDIT ===
>
> We get the results on Food-10. The mFID of the models with $K$=50, 500, 1000 is 60.8, 63.2, and 60.7. Similar to AnimalFaces-10, TUNIT performs reasonably well on Food-10. We added the results to Table 2. including density and coverage.
>
> **W2) Generalization performance on the choice of the classes**
>
> Thank you for your comment. Due to the time limit, it is highly likely that we could not finish the experiments during the rebuttal period. We will add the experimental results in the final version. Although it is not a direct answer, considering its performance on different datasets, we believe that our framework generalizes well against such data variations.
>
> **W3) Advanced clustering method**
>
> Please check the model E in Table 1.  We have reported the result using the state-of-the-art differential clustering method, invariant information clustering (IIC) [1]. As expected, the model with better clustering exhibits better performance compared to model C and D that use K-means.
>
> **W4) Set-level supervised TUNIT**
>
> Thank you for the good suggestion. We will add the performance of TUNIT with set-level supervision to Table 1 as soon as we get the full results. As a quick reference, we got the mFID of  45.2 for TUNIT with set-level supervision, which is better than the unsupervised TUNIT. We will add the results of density and coverage and results for Food-10 datasets as soon as we get those.
>
> **W5) Misuse of the word - supervised**
>
> Thank you. We reflected your suggestion in the paper.
>
> This is reference:
>
> [1] Ji, Xu, João F. Henriques, and Andrea Vedaldi. "Invariant information clustering for unsupervised image classification and segmentation." Proceedings of the IEEE International Conference on Computer Vision. 2019.

---

### Official Review · AnonReviewer3 · 2020-10-28
**The truly unsupervised image-to-image translation is interesting, and the idea to address it seems reasonable. The paper is generally well written. However, there are still some major issues in the motivation and the experiments.**

**Rating:** 5
**Confidence:** 3

**Review:**

In this work, the authors propose an approach for “truly” unsupervised image-to-image translation. Unlike the earlier image-to-image translation with image-level or set-level supervision, the proposed method introduces a contrastive learning method to simultaneously classify the image domain and transfer the domain without any supervision. Extensive experiments and ablation studies are conducted on labeled datasets (e.g. AnimalFaces and Food-101) and unlabeled datasets (e.g. FFHQ, LSUN car).

[Paper strengths]
- The setting of the "truly” unsupervised image-to-image translation is new (although might not be practical).
- The approach based on contrast learning to address this new problem is quite interesting and novel.
- In the experimentation, the claim about the automatic domain classification and translation is well demonstrated.

[Paper weakness]
1. Although the setting is new, it might not make sense in practice. For translation, no matter whether it is about language or image, we need to have target in mind. If no target is specified, translation does not make sense. Then, you are doing image generation, not translation. For example, for Fig. 3, if the source is a dog and the target is a dish, which is possible for the defined truly unsupervised scenario, then what will happen?
2. The main drawback of the experiments is lacking the fair comparisons to the SOTA. Although this work is an unsupervised method and the performance may have a gap to the SOTA (e.g. MUNIT, StarGAN, StarGAN v2), the authors should provide some examples to compare with the visual results or quantitative results of the SOTA.
3. The authors only compare the proposed method TUNIT with FUNIT in Tables 1, 3 and 4. The results are quite confusing. In Table 1, in general, the proposed unsupervised one TUNIT outperforms the fully supervised one FUNIT. Shouldn't FUNIT be the upper bound? Comparing TUNIT across Tables 1, 3 and 4, it seems having more ground-truth set labels does improve the performance at all, which is not reasonable.
4. In Figure 4, the visual results of FFHQ cannot preserve the identity of the source person. Is this meaningful? Does this a common issue in all methods? As the authors did not provide a fair comparison (even the supplement only shows the results of the proposed method), it’s hard to judge.
5. For Eq. (2), how do you choose positive and negative pairs? How to ensure the negative pairs hold different styles? I can't find the details.

----------
Update:
- Thanks the authors for the detailed reply. It addresses some of my concerns. Overall, I am fine if this paper gets accepted, given its novelty. My major concern is still with the experiment comparisons. The authors only compare the proposed method with FUNIT, which might not be fair since FUNIT is designed for few-shot translation, dealing with novel domains or classes with a few examples. The authors should compare with MUNIT or StarGANv2, with domain information obtained by clustering or any unsupervised methods.
- The authors should tune down the eye-catching statement of truly unsupervised translation since translating between dish and dog might not make sense, although this is a minor issue.

---

> ### Author Response · Authors · 2020-11-16
> **Official response to Reviewer #3 Part (2/2)**
>
> **W4) Identity preserving in FFHQ**
>
> The architecture for the general translation task suffers from preserving identity compared to the architecture for the facial attribute editing. For example, StarGANv2 conducts the translation task on AFHQ (animals) and CelebA-HQ (human). To preserve the identity of the input human face better than the basic model, StarGANv2 utilizes additional techniques such as high-pass filtering, skip connections, and even the face landmarks. Even though we believe that TUNIT can also be good at identity preservation by using additional techniques, we think that this is not the main focus of our work. Therefore, we do not apply the techniques for improving the identity preservation and do not conduct the fine-tuning on the hyperparameters for each dataset but use all the same hyperparameters and architectures to show the effectiveness of TUNIT on the unsupervised image-to-image translation task. Related to the results in Appendix, we add Appendix F, the comparative evaluation of FUNIT and TUNIT as Figure 4.
>
> **W5) About equation (2) and the negative samples**
>
> Thank you for pointing out this. It was dropped while we were trimming down our manuscript. As in MoCo[2], the positive sample is the augmented version of the query image (a real image from the dataset) and the negative samples are the augmented version of other images saved in a queue.
>
> Regarding the question “do the negative pairs really have different styles?”, the direct answer is that we do not guarantee the negative pairs to really have different styles (because no style definition is given in this problem). Nevertheless, TUNIT successfully learns the meaningful representation to distinguish various styles and achieves high quality results. Similarly, existing unsupervised contrastive learning methods also show that their trained network can successfully conduct downstream tasks such as classification and detection via finetuning although they do not guarantee negative pairs are in different classes. This intriguing result has been reported and investigated in the previous studies [2, 3, 4]. In [4], the authors conducted a theoretical analysis of contrastive learning in terms of the negative samples. They showed that the loss where a query and a negative sample have the same class is controlled by the intraclass deviation of a representation learner (function) and the loss of the downstream supervised task is bounded by the loss of the unsupervised representation task. That is, if we use sufficient samples and a large queue to train the network, the intraclass deviation decreases so that still the network is able to conduct the downstream task. We add the citation.
> In summary, although all unsupervised contrastive learning methods might suffer from the uncertainty of negative pairs (they might not have really negative styles or classes), the common practice of choosing the augmented version of different samples has been shown successfully in various downstream tasks. To be clear, we add the description for equation (2) as “x and x$^+$ denote an image and randomly augmented version of x, respectively”
>
> These are references:
>
> [1] Chen, Ting, et al. "Big self-supervised models are strong semi-supervised learners." Advances in Neural Information Processing Systems 33 (2020).
>
> [2] He, Kaiming, et al. "Momentum contrast for unsupervised visual representation learning." Proceedings of the IEEE/CVF Conference on Computer Vision and Pattern Recognition. 2020.
>
> [3] Chen, Ting, et al. "A simple framework for contrastive learning of visual representations." arXiv preprint arXiv:2002.05709 (2020).
>
> [4] Saunshi, Nikunj, et al. "A theoretical analysis of contrastive unsupervised representation learning." International Conference on Machine Learning. 2019.

---

> ### Author Response · Authors · 2020-11-16
> **Official response to Reviewer #3 Part (1/2)**
>
> We thank the reviewer3 for the constructive comment on our work. We are encouraged to hear that the problem is new, the approach is interesting, and our experimental claim is well demonstrated. Here we address the five main concerns:
>
> **W1) Practical significance of TUNIT**
>
> In fact, there are many cases where a dataset contains a vague definition of domains. Domains in a set of particular objects can be divided by many different attributes, e.g., glasses, gender, and hairstyle on human faces. or the color and shape of cars, and collecting labels of all samples is laborious. Even in such cases, TUNIT can conduct the translation and also provide a guide for defining domains in the dataset. This property of TUNIT has significance in terms of research.
> The representative examples of dealing with unlabeled datasets are unsupervised clustering and representation learning. Even in clustering, no target class is given, but by itself, it provides insightful results to the researchers and is widely adopted for practical usage. And a network trained with unsupervised representation learning solely has no meaning itself. It is just a set of trained weights. However, if we use the feature extractor with a clustering algorithm, we can group the images having similar semantics, or the trained weight can be a strong classifier with a small number of labeled samples [1].
> In section 3.3, we train TUNIT with 1% of labeled samples for guiding the domain separation. We note that 1% is around ten images in each domain for AnimalFaces-10 and Food-10. Even if we use the small number of labeled samples, TUNIT still works well as we intended. In this point of view, we believe that TUNIT has practical significance.
>
> **W2) Baselines**
>
> We also considered the suggested models as baselines but we eventually decided to use FUNIT as our baseline due to the following reasons. Considering that our aim is to conduct multi-domain image-to-image translation, MUNIT is not a proper baseline. Not only does it require set-level supervision, but it requires to train a combinatorial number of models to perform multi-domain image translation. More specifically, we can apply MUNIT to conduct the ten-domain translation by training 55 models, which is overwhelming. While StarGAN can handle multi-domain translation, it is not state-of-the-art. Then, there remain two candidates -- FUNIT and StarGANv2. Between these two, StarGANv2 still requires full set-level supervision while FUNIT requires less. FUNIT reduces the amount of supervision at the test phase by adopting the few-shot learning scheme. In terms of the level of supervision and the relevance to our main task, we believe that FUNIT is the most proper baseline among the state-of-the-art models of the field.
>
> **W3) Shouldn't FUNIT be the upper bound? The unsupervised model seems to outperform the set-level supervised models.**
>
> If we understood the reviewer’s intention correctly, the reviewer is asking how and why the model G (TUNIT, unsupervised) in Table 1 can show better performance than the model B (FUNIT framework but our improved model architecture, set-level supervised). In general, this question is valid since supervised performance is the upper bound of unsupervised one, which seems not to be the case here.
> This seemingly impossible result is in fact due to a set of loss functions and a guiding network that we proposed for TUNIT, which are missing in the model B. To fairly study a remaining gap between the supervised and the unsupervised methods, we must compare the model with the same architecture and the loss functions, except the existence of supervision.
> Given the entire true domain labels, the supervised TUNIT model achieves mFID of 45.2 on AnimalFaces-10, which is indeed better than the unsupervised one (47.7). Thus, there is still the gap. For readers to better understand, we will add the full result in Table 1 as soon as we finish the experiments.
>
> (cont.)

---

### Comment · Area_Chair1 · 2020-11-21
**The discussion stage is open**

The discussion stage is open

Dear Reviewers:

Thanks for your insightful reviews! Now the discussion stage is open and the authors have posted their responses. We will appreciate that the following things-to-do can be done by Tues, Nov 24.

1 Acknowledge explicitly that you have read the responses.

2 Modify your review if necessary.

3 Communicate with the authors/reviewers/AC by adding/responding to the comments if necessary.

Thanks a lot!

---

### Decision · Program_Chairs · 2021-01-07
**Final Decision**

**Decision:**

Reject

**Comment:**

This paper defines a truly unsupervised image translation scenario. Namely, there are no parallel images or domain labels. To achieve robust performance in this scenario, the authors use 1) clustering and 2) generator-discriminator structure to map images from different domains and generate images for target domains.

In all, all the reviewers agree that this definition of unsupervised image translation is interesting. However, there are also several concerns for the real-world practical application and empirical results.  Unlike unsupervised text translation whose target language is known, the truly unsupervised image translation is difficult to make sense without identifying what is the target domain. This limits the contribution of this paper to some specific tasks instead of more general tasks. For the empirical results, the selection of data and the hyperparameter K do not convince the reviewers.